# Nonredox trivalent nickel catalyzing nucleophilic electrooxidation of organics

Yuandong Yan[1], Ruyi Wang[1], Qian Zheng[1], Jiaying Zhong[1], Weichang Hao [2], Shicheng Yan [1,3] ✉ & Zhigang Zou [1,3]

A thorough comprehension of the mechanism behind organic electrooxidation is crucial for the development of efficient energy conversion technology. Here, we find that trivalent nickel is capable of oxidizing organics through a nucleophilic attack and electron transfer via a nonredox process. This nonredox trivalent nickel exhibits exceptional kinetic efficiency in oxidizing organics that possess the highest occupied molecular orbital energy levels ranging from $-7.4$ to $-6$ eV (vs. Vacuum level) and the dual local softness values of nucleophilic atoms in nucleophilic functional groups, such as hydroxyls (methanol, ethanol, benzyl alcohol), carbonyls (formamide, urea, formaldehyde, glucose, and N-acetyl glucosamine), and aminos (benzylamine), ranging from $-0.65$ to $-0.15$. The rapid electrooxidation kinetics can be attributed to the isoenergetic channels created by the nucleophilic attack and the nonredox electron transfer via the unoccupied $e_g$ orbitals of trivalent nickel ($t_{2g}^6 e_g^1$). Our findings are valuable in identifying kinetically fast organic electrooxidation on nonredox catalysts for efficient energy conversions.

Hydrogen, as an energy carrier, will play a crucial role in the energy revolution and decarbonization of multiple industries[1]. In particular, electrochemical hydrogen production is an attractive technique for meeting the needs of energy conservation, emission reduction, and carbon neutrality. It is a key factor in ensuring a robust supply of renewable energy, as hydrogen is capable of long-term storage and long-range transport, overcoming the intrinsic disadvantages of intermittent and unevenly distributed renewables[2]. Hydrogen can be generated through the electrocatalysis of the earth's abundant biomass resources (such as alcohol, glucose, and lignin) and industrial wastewater (such as urea and hydrazine)[3]. More importantly, electrolysis of most organics presents a much lower thermodynamic potential than high-energy-consuming water oxidation. The thermodynamic potential of the oxygen evolution reaction (OER) of water splitting is typically 1.23 V, which is far higher than those of the urea oxidation reaction (UOR, 0.37 V), hydrazine oxidation reaction (HzOR, 0.33 V), ammonia oxidation reaction (AOR, 0.06 V), and glucose oxidation

reaction (0.05 V)[4–6]. Therefore, replacing the water-splitting OER with thermodynamically favorable electrooxidation of organics offers an energy-saving strategy that is useful in producing high-value-added chemicals or treating industrial wastewater[7].

Ni-based electrocatalysts are widely used in large-scale industrial water electrolysis[8,9] and are also efficient in electrooxidation of organics[10–13]. However, there is a big difference in electrochemical processes for water oxidation and organics oxidation on Ni-based catalysts. As shown in Fig. 1a, for the Ni(OH)$_2$ electrode, the water oxidation usually occurs after the Ni$^{2+}$(OH)$_2$/Ni$^{3+}$OOH oxidation and their potential gap is significant enough to use Ni(OH)$_2$/NiOOH as a redox couple to create the decoupling OER and hydrogen evolution reaction (HER)[14]. This means that although oxidizing Ni$^{2+}$ (3d$^8$, $t_{2g}^6 e_g^2$) to Ni$^{3+}$ (3d$^7$, $t_{2g}^6 e_g^1$) via Ni(OH)$_2$ → NiOOH + H$^+$ + e$^-$, a proton-coupled electron transfer (PCET) process, offers the unoccupied orbitals to transfer electrons[15], the NiOOH is still unable to oxidize water, unlike the higher-valence Ni$^{δ+}$ ($δ > 3$). In contrast, totally different from the

[1]Collaborative Innovation Center of Advanced Microstructures, National Laboratory of Solid State Microstructures, College of Engineering and Applied Sciences, Nanjing University, No. 22 Hankou Road, Nanjing, Jiangsu 210093, China. [2]School of Physics, Beihang University, 37 Xueyuan Road, 100191 Beijing, China. [3]Jiangsu Key Laboratory for Nano Technology, Eco-materials and Renewable Energy Research Center (ERERC), Nanjing University, No. 22 Hankou Road, Nanjing, Jiangsu 210093, China. ✉e-mail: yscfei@nju.edu.cn

OER process, the organics (Fig. 1b), such as alcohols, glucose, urea, and hydrazine, are immediately electrolyzed once the Ni(OH)$_2$ is oxidized to NiOOH[10–13]. However, there is a lack of knowledge about the nature for the big difference in oxidizing water and organics on Ni-based catalysts and the catalytic mechanism reminds unclear and is under debate, greatly hindering the discovery of efficient catalysts.

In this study, as shown in Fig. 1c, using the various theoretical and experimental techniques, we clearly show that the electrooxidation of organics on NiOOH does not follow the Ni(OH)$_2$/NiOOH redox-mediated electron transfer mechanism, a popular viewpoint that was suggested in most reports[12,15], but rather a nonredox electron transfer process without oxidation state change of Ni$^{3+}$ species. The trivalent nickel acts as electrophilic electrooxidation center and creates an isoenergetic transfer channel with organics possessing the highest occupied molecular orbital (HOMO) energy levels ranging from −7.4 to −6 eV (vs. Vacuum level) and the dual local softness ($\Delta s_k$) values of nucleophilic atoms in nucleophilic functional groups, such as hydroxyls (methanol, ethanol, benzyl alcohol), carbonyls (formamide, urea, formaldehyde, glucose, and N-acetyl glucosamine), and aminos (benzylamine), with values ranging from −0.65 to −0.15. The (HOMO, $\Delta s_k$) combination criterion can well explain why the Ni$^{3+}$ does not oxidize water efficiently, but it works for electrooxidation of organics. The rapid electrooxidation kinetics of organics can be attributed to the isoenergetic channels created by the nucleophilic attack and nonredox electron transfer via the unoccupied e$_g$ orbitals of trivalent nickel (t$_{2g}^6$e$_g^1$). The (HOMO, $\Delta s_k$) combination criterion is valuable in

identifying kinetically fast organic electrooxidation on nonredox catalysts for efficient energy conversions.

## Results

### OER and UOR on Ni(OH)$_2$ electrode

To explore the origins of Ni catalyzing electrooxidation of organics, the urea oxidation on Ni(OH)$_2$ electrode was selected as a model reaction (Supplementary Fig. 1). We first compare the electrooxidation of water and urea on Ni(OH)$_2$ electrode. As shown in Fig. 2a, linear sweep voltammetry (LSV) OER polarization curve in 1 M KOH electrolyte clearly exhibited an initial Ni(OH)$_2$/NiOOH oxidation wave at 1.32 V and subsequent OER current at 1.55 V. The big potential gap between Ni(OH)$_2$/NiOOH oxidation and water oxidation implies that occurrence of water oxidation requires polarizing the electrode to the higher-valence Ni$^{\delta+}$ ($\delta > 3$) species (Fig. 2a and Supplementary Fig. 2). In contrast, the UOR current in 0.33 M urea + 1 M KOH electrolyte is immediately detected when initially oxidizing Ni(OH)$_2$ to NiOOH, suggesting no kinetic delay for Ni$^{3+}$ oxidizing urea. The cyclic voltammetry (CV) at a slow scan rate of 1 mV s$^{-1}$ revealed a reversible Ni(OH)$_2$/NiOOH interconversion when polarizing Ni(OH)$_2$ electrode in 1 M KOH electrolyte (Inset in Fig. 2a), but no Ni(OH)$_2$/NiOOH redox process can be decoupled from the CV curve for polarizing Ni(OH)$_2$ electrode in 0.33 M urea + 1 M KOH electrolyte. The UOR current begins at 1.32 V, a potential for oxidizing of Ni(OH)$_2$ to NiOOH, during positive-going potential sweep and no reduction wave of NiOOH to Ni(OH)$_2$ is observed during negative-going potential sweep, probably suggesting

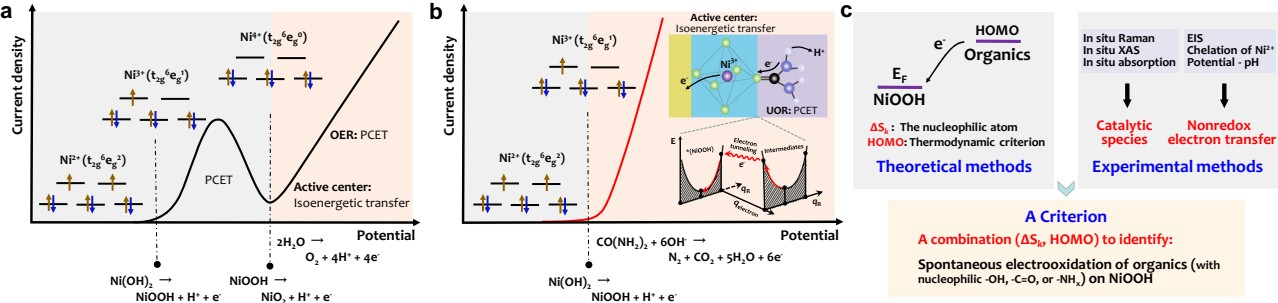

**Fig. 1 | The challenges in understanding UOR and OER on Ni(OH)$_2$ electrode. a** A typical anodic polarization curve for water electrooxidation on Ni(OH)$_2$ electrode in 1 M KOH electrolyte. The Ni$^{2+}$ (t$_{2g}^6$e$_g^2$) is in turn oxidized to Ni$^{3+}$ (t$_{2g}^6$e$_g^1$), and to Ni$^{\delta+}$ ($\delta > 3$, t$_{2g}^6$e$_g^0$ for Ni$^{4+}$) via a proton-coupled electron transfer (PCET) process. An isoenergetic electron transfer is created between Ni$^{\delta+}$ ($\delta > 3$) and oxygen-containing intermediates ($^\cdot$OH, $^\cdot$O, $^\cdot$OOH) of the elemental reactions during oxygen evolution reaction (OER). The brown and blue arrows correspond to spin-up and spin-down electrons, respectively. **b** A typical anodic polarization curve for urea oxidation

reaction (UOR) on Ni(OH)$_2$ electrode. The UOR current initials at the Ni$^{2+/3+}$ oxidation potential, implying that it is possible to create an isoenergetic electron transfer at each elemental reaction step during Ni$^{3+}$ oxidizing urea (Inset in Fig. 1b). Insets show the electron transfer mechanism. $q_R$ is nuclear coordinate, $q_{electron}$ is electron coordinate, $^\cdot$(NiOOH) is active center. **c** A scheme to describe our strategy to discover the electrooxidation mechanism of organics on Ni$^{3+}$ active center with a nonredox electron transfer.

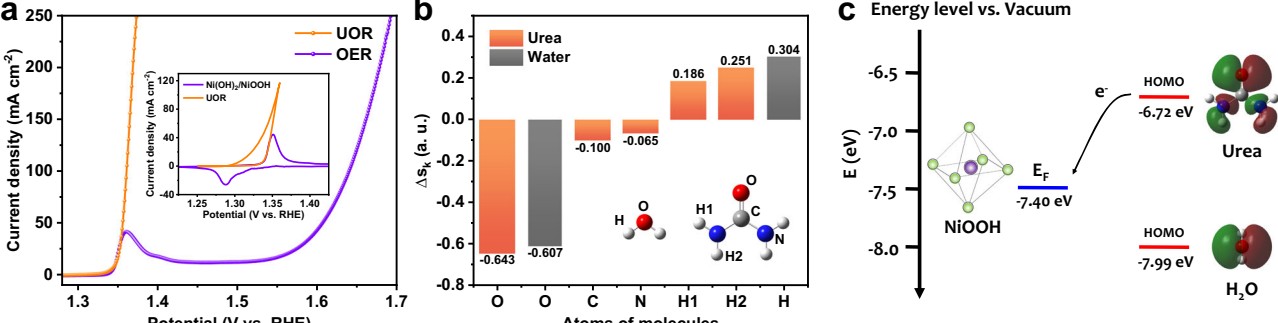

**Fig. 2 | The intrinsic differences between UOR and OER. a** The UOR and OER polarization curves on Ni(OH)$_2$ electrode. The inset shows the CV scans for Ni(OH)$_2$ electrode in 1 M KOH or 0.33 M urea + 1 M KOH in the Ni(OH)$_2$/NiOOH redox

potential region. **b** The calculated $\Delta s_k$ values for atoms in urea and water to identify the nucleophilic atoms. **c** The HOMO levels of urea and water and the Fermi level of NiOOH.

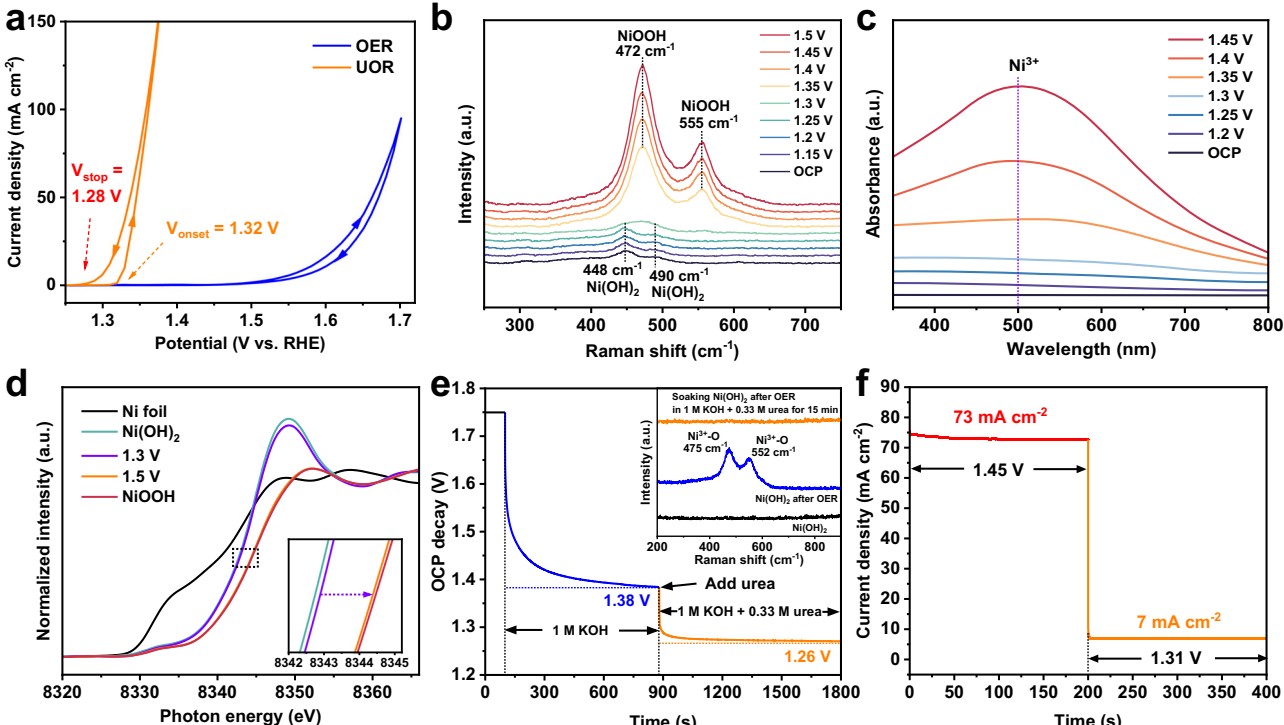

**Fig. 3 | Evidences to trivalent nickel as active species for urea electrooxidation. a** UOR and OER LSV curves derived by fitting the chronoamperometric current-potential data. **b** Potential-dependent Raman spectra for polarizing Ni(OH)$_2$ electrode in 1 M KOH + 0.33 M urea. **c** Potential-dependent UV-vis absorption spectra for Ni(OH)$_2$ electrode in 1 M KOH + 0.33 M urea. **d** Normalized in situ Ni K-edge XANES spectra for polarizing Ni(OH)$_2$ electrode at different potentials in 1 M KOH + 0.33 M urea, referenced to Ni foil and NiOOH. **e** After polarizing Ni(OH)$_2$ electrode at 70 mA cm$^{-2}$, the $V_{OCP}$ decay was in turn observed in 1 M KOH and in 1 M KOH + 0.33 M urea. The inset is Raman spectra for the as-prepared Ni(OH)$_2$, the Ni(OH)$_2$ after OER, and the Ni(OH)$_2$ after OER with soaking in 1 M KOH + 0.33 M urea for 15 min. **f** The i-t curve of the Ni (OH)$_2$ electrode when abruptly switching potentials from 1.45 to 1.31 V in 1 M KOH + 0.33 M urea.

that the Ni$^{3+}$ oxidizing urea does not follow the electron transfer mechanism of Ni(OH)$_2$/NiOOH redox couple. The findings suggest that understanding the UOR and OER mechanism on the Ni(OH)$_2$ electrode is crucial for elucidating the Ni catalyzing electrooxidation of organics.

First, we conducted theoretical studies to understand the catalysis of OER and UOR on the Ni(OH)$_2$ electrode. In principle, the electro-oxidation of molecules is triggered by initial adsorption and subsequent energy transfer. An initial adsorption equilibrium state is established by orbital overlap between the catalytic center and reagents, creating energy transfer channels, the bonding orbitals for electron transfer of oxidation reactions[16]. The first goal is to understand the interactions between urea and Ni$^{3+}$ active species and the thermodynamics of electron transfer during UOR. We accordingly check the dual local softness, $\Delta s_k$, and the highest-occupied molecular orbital (HOMO) level of urea and the Fermi level of NiOOH, the real active species for UOR. The dual local softness[17], $\Delta s_k$, is defined as the condensed dual descriptor of $\Delta f_k$ multiplied by the molecular softness and is useful to identify which sites in a molecule are more favorable for a nucleophile, or an electrophile attack[18,19]. A most negative $\Delta s_k$ value, −0.643, for carbonyl oxygen in urea suggests its nucleophilic nature, which tends to attack the coordinately unsaturated Ni$^{3+}$ site of NiOOH (Fig. 2b). This is because the coordinatively unsaturated Ni sites in NiOOH are positively charged, an electrophilic center, and seem to be the most possible sites that bear the nucleophilic attack. The coordinatively unsaturated Ni sites usually are edge sites or oxygen vacancies (O$_{Vs}$)[20,21]. We accordingly created the NiOOH (100) surface slab with O$_{Vs}$, a highly active facet with coordinately unsaturated Ni sites[22,23], for theoretical calculations (Supplementary Fig. 3). The first-principles calculations indicated that the Fermi level (−7.4 eV vs. Vaccum level) of NiOOH is more negative than the HOMO level (−6.72 eV) of urea (Fig. 2c). This implied that the chemical potential for electrons

at HOMO level of urea is high enough to drive the electron transfer from bonding orbitals to catalytic center during UOR. In contrast, although the $\Delta s_k$ value of oxygen in water molecule is significantly negative to be -0.607, the HOMO level of water at -7.99 eV, is deeper than the Fermi level of NiOOH, suggesting the higher thermodynamic requirement for electron transfer to trigger water oxidation. The big energy difference between the HOMO orbital of water and Fermi level of NiOOH revealed their weak interactions due to the difficulty in electron exchange to build an initial adsorbed equilibrium state, thus resulting in the inefficient water oxidation on Ni$^{3+}$ species. On the basis of theoretical and experimental results for the OER and UOR on Ni$^{3+}$ species, we can conclude that a combination of the $\Delta s_k$ value and HOMO level can well explain why the Ni$^{3+}$ does not oxidize water efficiently, but it works for urea electrooxidation.

## Evidences to UOR driven by Ni$^{3+}$

The chronoamperometric steps method was carried out to detect the real UOR current on Ni(OH)$_2$ electrode without the unsteady-state currents, resulting from the capacitive charging effect and/or oxidation of intermediates[24]. To obtain a complete static current-potential curve, the steady-state current-potential data were recorded by varying potentials with an increment of 10 mV. Indeed, as indicated by static OER current-potential data (Fig. 3a), chronoamperometric steps method can completely eliminate the unsteady-state currents, such as the oxidation or reduction of Ni species[24]. The positive- and negative-going OER CV branches share the same onset potential to be at 1.55 V, undoubtedly evidencing that the Ni$^{3+}$ has no ability to oxidize water. However, the onset potential of UOR in positive-going potential branch is 1.32 V, which is 40 mV higher than 1.28 V in the negative-going one. When positively going the potentials, the low-conductivity Ni(OH)$_2$ is gradually discharging to meet the potential requirement for

generation of high-conductivity NiOOH[25], that is, the electrode changes from low-valence low-conductivity to high-valence high-conductivity state. The 1.32 V potential requirement for initially oxidizing $Ni(OH)_2$ to NiOOH is attributed to the inherently low conductivity of $Ni(OH)_2$. In contrast, as negatively going the potentials, the electrode changes from high- to low-conductivity state, thus exhibiting a lower UOR onset potential at 1.28 V, a critical potential for completely reducing NiOOH to $Ni(OH)_2$. The UOR onset potential as low as 1.28 V is a direct indicator to confirm that the $Ni^{3+}$ is capable of directly oxidizing urea. In addition, at the near onset potential, the pure UOR current can be observed by chronoamperometric steps method and the coupled currents of both $Ni(OH)_2$/NiOOH oxidation and UOR were recorded by the CV scan with a slow scan rate of $1\,mV\,s^{-1}$. However, the initial currents observed in the two methods share the same UOR onset potential, well indicating that the urea oxidation immediately occurs once the $Ni(OH)_2$ is oxidized to NiOOH.

To visualize the existence of $Ni^{3+}$ species during UOR, the potential-dependent Raman spectra were recorded in 0.33 M urea + 1 M KOH electrolyte (Fig. 3b and Supplementary Fig. 4). Two characteristic Raman bands for as-prepared $Ni(OH)_2$ are $448\,cm^{-1}$ for Ni-OH symmetric stretching vibration and $490\,cm^{-1}$ for Ni-O vibration[26]. Obviously, a depolarized $E_g$ mode at $472\,cm^{-1}$ and a polarized $A_{1g}$ mode at $555\,cm^{-1}$ for the $NiO_6$ units in the $[NiO_2]$ framework of NiOOH occur at potentials above 1.35 V[27], confirming that the NiOOH stably exists during UOR. And the light absorption of $Ni^{3+}$ species at 500 nm was observed in situ electrochemical UV-vis absorption spectra at potentials above 1.3 V[28] (Fig. 3c and Supplementary Fig. 5). The in situ Ni K-edge X-ray absorption near edge structure (XANES) spectra revealed that the intensity of the white line decreases with polarizing the $Ni(OH)_2$ electrode from 1.3 to 1.5 V in 1 M KOH + 0.33 M urea (Fig. 3d), suggesting the formation of higher-valence Ni species due to that the high-valence Ni inducing distortion in the $NiO_6$ octahedral configuration will result in a decrease in the white line intensity[29]. Meanwhile, the edge energy (measured at half height) for the electrode at 1.3 and 1.5 V is the same as those of the $Ni(OH)_2$ (8342.6 eV) and NiOOH (8344.5 eV), respectively. This edge position suggests a +3 nickel oxidation state for the electrode during UOR[30]. The in situ XANES results also confirmed that during UOR the $Ni^{3+}$ is UOR-active and is likely to keep a constant valence state, in good agreement with the results from in situ Raman and UV-vis absorption. These evidences confirmed that the urea is oxidized by $Ni^{3+}$ species. In principle, the open circuit potential ($V_{OCP}$) describes the quasi-Fermi level ($E_{electrode}$) of the electrode against reference electrode when the electrode is under open-circuit conditions[31]. In our case, the $V_{OCP}$ decay is a result of the $E_{electrode}$ shift, equivalent to changes in the average oxidation state of active Ni species. Therefore, the $V_{OCP}$ decay provides a route to monitor the reaction kinetics between $Ni^{3+}$ and urea. The $V_{OCP}$ decay measurements were carried out after chronopotentiometry at an OER current of $70\,mA\,cm^{-2}$ in 1 M KOH electrolyte to polarize the Ni species at high-valence states. During this period, the electrode potential reached its steady-state value and remained constant. After turning off the electricity input, the $V_{OCP}$ decay was immediately monitored and stopped at a quasi-equilibrium potential of 1.38 V (Fig. 3e), a potential for $Ni^{3+}$ generation (Fig. 2a). The $V_{OCP}$ decay is a result of reducing high-valence Ni species by the oxidative species from electrolyte to upshift the $E_{electrode}$. The quasi-equilibrium potential is indicative of the small energy difference between $E_{electrode}$ and water oxidation energy level ($E_{O_2/OH^-}$), which is not enough to overcome the barriers of water oxidation by active Ni species to occur. Adding urea will trigger the sharply second $V_{OCP}$ decay from 1.38 to 1.26 V, a potential for $Ni^{2+}$ generation[32], clearly indicating that the urea oxidation by $Ni^{3+}$ is a spontaneous chemical reaction with rapid kinetics. Correspondingly, the Raman signals of NiOOH can be detected after polarizing $Ni(OH)_2$ electrode in 1 M KOH and completely vanished when soaking the electrode in 0.33 M urea + 1 M KOH electrolyte, demonstrating that the

applied potentials are in charge of maintaining the highly active $Ni^{3+}$ species for UOR to proceed (in set in Fig. 3e). A $73\,mA\,cm^{-2}$ UOR current was immediately switched to steady-state $7\,mA\,cm^{-2}$ when abruptly altering the electrode potential from 1.45 to 1.31 V (Fig. 3f and Supplementary Fig. 6), revealing that the urea oxidation is highly sensitive to the amount of $Ni^{3+}$ and the electron transfer during UOR is extremely rapid to create new equilibrium state.

### Nonredox $Ni^{3+}$ catalyzing UOR

Next, we explore the electron transfer during $Ni^{3+}$ oxidizing urea. The kinetic differences between urea oxidation and $Ni(OH)_2$/NiOOH oxidation were first visualized by dual electrochemical workstation with a purpose of simultaneously monitoring the anodic potentials of working electrode and the cell voltage (Fig. 4a and Supplementary Fig. 7). After confirming that the Pt electrode as the cathode is able to minimize the effects of cathodic kinetics on cell voltage (Supplementary Fig. 7a, b), the kinetics of the $Ni(OH)_2$ anode was checked during UOR. We periodically alter the anodic potentials between 1.4 and 1.45 V with a stay time of 10 s at every potential point, the cell voltage is real-timely monitored (Fig. 4b and Supplementary Fig. 7c). According to the potential window of $Ni(OH)_2$/NiOOH oxidation (Fig. 2a), the 1.4 and 1.45 V are high enough to completely oxidize $Ni(OH)_2$ to NiOOH, thus providing a route to detect the oxidation reaction kinetics. Abruptly imposing 1.45 V on the as-prepared $Ni(OH)_2$, the cell voltage to reach an equilibrium state is a time-consuming process due to the kinetically sluggish oxidation of $Ni(OH)_2$ to NiOOH. Completely oxidizing of $Ni(OH)_2$ likes to be achieved at the seventh potential rising stage from 1.4 to 1.45 V, proving that the electron transfer during $Ni(OH)_2$/NiOOH oxidation is high-barrier process. In contrast, at every potential decreasing stage from 1.45 to 1.4 V, the cell voltage immediately reaches the equilibrium state, indicating the most likely path is the $Ni^{3+}$ reacting with urea without the slow time-consuming $Ni(OH)_2$/NiOOH oxidation process, to rapidly adjust the average valence states of the electrode, thus sensitively adapting to the potential changes. Accordingly, we can conclude that the UOR is directly driven by $Ni^{3+}$ without the $Ni(OH)_2$/NiOOH redox process to transfer electrons. To further prove this fact, the dimethylglyoxime disodium salt octahydrate ($C_4H_6N_2Na_2O_2 \cdot 8H_2O$) was utilized as a probe molecule to verify the electron transfer mechanism because dimethylglyoxime is a chelating agent which can form wine-red soluble complexes with $Ni^{2+}$ in alkaline media[33,34]. In 20 mM $C_4H_8N_2Na_2O_2$ + 0.33 M urea + 1 M KOH, no color changes were visible during constant voltage testing at 1.55 V for 6 h and subsequently switching potential to 1.31 V for 24 h (Fig. 4c and Supplementary Fig. 8), strongly suggesting no interconversion of $Ni(OH)_2$/NiOOH on the electrode during UOR. This fact supports a standpoint that urea oxidation by NiOOH is probably achieved by direct electron transfer via $Ni^{3+}$ catalytic center. In the presence of dimethylglyoxime, we performed 2400 CV scans between 0.9 and 1.6 V to simulate a situation of the $Ni(OH)_2$/NiOOH interconversion during urea oxidation (Supplementary Fig. 8c). After 2400 CV scans, the $Ni(OH)_2$ was completely removed and the electrode exhibited low UOR activity (Fig. 4d), demonstrating the strong chelating ability between dimethylglyoxime and $Ni^{2+}$. Indeed, the wine-red complex occurs during the CV scans (Inset in Fig. 4d), well verifying that if there is a $Ni(OH)_2$/NiOOH interconversion to transfer electrons the complexing reaction of dimethylglyoxime with $Ni^{2+}$ necessarily occurs. The complexing reaction was also occurring after spontaneous chemical reduction of $Ni^{3+}$ by urea under open-circuit conditions (Supplementary Fig. 9). These evidences well rule out a possibility of transferring electrons by $Ni(OH)_2$/NiOOH redox couple during UOR. The activity of $Ni^{3+}$ is significantly enough to chemically oxidize urea, making us believe that the applied potentials above $Ni(OH)_2$/NiOOH oxidation potential are mainly responsive for extracting electrons that are received from urea into NiOOH. Therefore, under applied potentials above $Ni(OH)_2$/NiOOH oxidation potential, the electron injection from

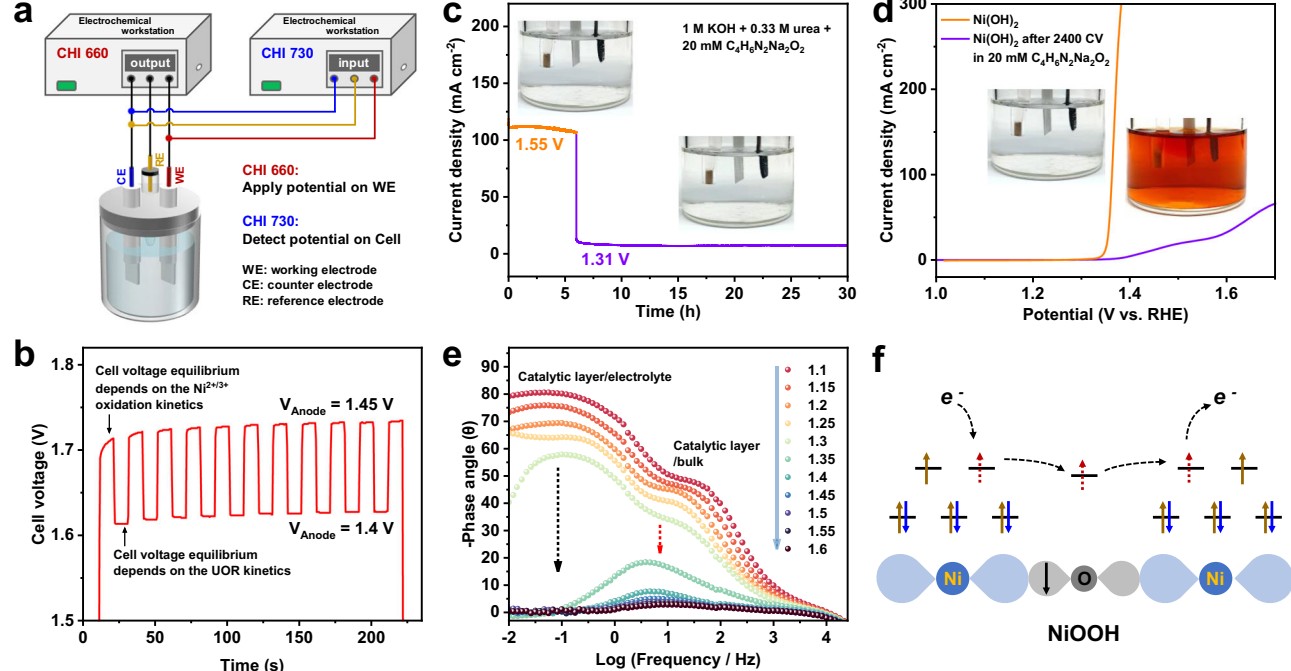

**Fig. 4 | The nonredox electron transfer on trivalent nickel. a** A scheme to show the operating principle of dual electrochemical workstation to simultaneously monitor the anodic potentials of working electrode and the cell voltage. **b** The time-dependent cell voltage equilibrium when periodically altering the anodic potentials between 1.4 and 1.45 V. **c** UOR at 1.55 V for 6 h and 1.31 V for 24 h in 20 mM $C_4H_8N_2Na_2O_2$ + 0.33 M urea + 1 M KOH. Insets show the optical photos of electrolytes. **d** The LSV polarization curve for $Ni(OH)_2$ electrode after 2400 CV scans in 20 mM $C_4H_8N_2Na_2O_2$ + 0.33 M urea + 1 M KOH. Insets show the color changes of electrolytes. **e** Potential-dependent EIS Bode plots of $Ni(OH)_2$ electrode when positive-going potentials varied from 1.1 to 1.6 V in 0.33 M urea + 1 M KOH. **f** The nonredox electron transfer via the unoccupied $e_g$ orbitals of $Ni^{3+}$ ($t_{2g}^6 e_g^1$) in Ni-O-Ni configuration of NiOOH.

urea to unoccupied orbitals of $Ni^{3+}$ can be directly extracted into external circuit with no need of valence variation of $Ni^{3+}$ species.

The fast electron transfer during UOR on $Ni(OH)_2$ was further confirmed by electrochemical impedance spectroscopy (EIS). We combine the potential-dependent Nyquist and Bode plots to accurately assign the electron transfer during UOR. Bode plots described two electrochemical processes on $Ni(OH)_2$ electrode (Fig. 4e and Supplementary Fig. 10), respectively exhibiting phase angle at high-frequency region (≥1 Hz) for electron transfer at the interface between the catalytic layer and the bulk of catalyst and the phase angle at low-frequency region (<1 Hz) for electron transfer from the UOR intermediates to active species. As shown in Supplementary Fig. 10a, b, in low-frequency region (<1 Hz), Bode plots of $Ni(OH)_2$ electrode exhibit a phase angle when varying positive-going potentials from 1.1 to 1.3 V, which is assigned to the discharging process at interface between $Ni(OH)_2$ and electrolyte. However, at potentials positive than 1.3 V, the disappearance of phase angle in low-frequency region is an indication of surface reconstruction, $Ni(OH)_2$ to NiOOH, and simultaneously UOR to occur on the NiOOH active layer with isoenergetic channels between NiOOH and urea, in good agreement with UOR onset potential determined by LSV result (Fig. 3a). This validates that the electron transfer is a nearly barrier-free process for UOR on NiOOH, corresponding to fast decrease of semicircle in Nyquist plots at potentials positive than 1.3 V (Supplementary Fig. 10c, d). Correspondingly, in high-frequency region (≥1 Hz), the phase angles describe the electron transfer at the interface between the catalyst bulk and the surface active layer. When potentials are positive than 1.3 V, a fast potential-dependent decrease in high-frequency phase angle verifies the initial oxidation of low-conductivity $Ni(OH)_2$ to high-conductivity NiOOH. The high-frequency phase angle sharply decreased and shifted forward higher frequency with extending the UOR potential window, demonstrating that the electron transfer is low-barrier process from surface catalytic layer to NiOOH bulk. At 1.35 V UOR potential, the smaller phase angle in

Bode plots and the smaller semicircle in Nyquist plots for negative-going process than positive-going process would suggest that the high-conductivity NiOOH is the UOR active species (Supplementary Figs. 10, 11). Fitting the Nyquist plot by a typical Randle's circuit (Supplementary Fig. 12), both the electron transfer resistances at the catalyst bulk/catalytic layer interface ($R_{ct,M}$) and the catalytic layer/electrolyte interface ($R_{ct,UOR}$) are as low as 0.076 - 0.487 ohm when the applied potential is higher than 1.30 V, a UOR potential, suggesting that the electron transfer during UOR on NiOOH is similar to the electron conduction in a metallic conductor. The femtosecond transient absorption spectra (fs-TAS) were carried out to provide experimental evidence to show the electron transfer kinetics in the $Ni^{3+}$ species in a femtosecond time scale. The excited-state $Ni(OH)_2$ with and without adsorption of urea (denoted as urea-$Ni(OH)_2$) was adopted to describe the $Ni^{3+}$ state and the $Ni^{3+}$ state with electron accumulation due to that the urea with nucleophilic carbonyls is an electron donor (Supplementary Fig. 13a, b). After light excitation at 320 nm, the decay time from excited state of $Ni(OH)_2$ and urea-$Ni(OH)_2$ to ground state is 8 ps and 15 ps, respectively, indicating the charge transfer on $Ni^{3+}$ state is faster than that on $Ni^{3+}$ state with electron accumulation. Thus, electrons tend to transfer through $Ni^{3+}$ directly during UOR, rather than accumulating to form transient $Ni^{2+}$ which hinders electron transfer. The $Ni^{3+}$ oxidizing urea at each elemental reaction step would follow an isoenergetic electron transfer mechanism, a direct electron transfer via the unoccupied $e_g$ orbitals of $Ni^{3+}$ ($t_{2g}^6 e_g^1$) in Ni-O-Ni configuration without the $Ni(OH)_2$/NiOOH redox process, inherently possessing faster transfer kinetics (Fig. 4f).

### The origins of $Ni^{3+}$ catalyzing UOR
The electron transfer amounts for $N_2$ generation, with a faradaic efficiency of 94.5% at potentials above 1.4 V (Supplementary Fig. 14a), were equal to the amount of electric charge which has passed through the electrode during UOR, suggesting a UOR catalytic reaction to

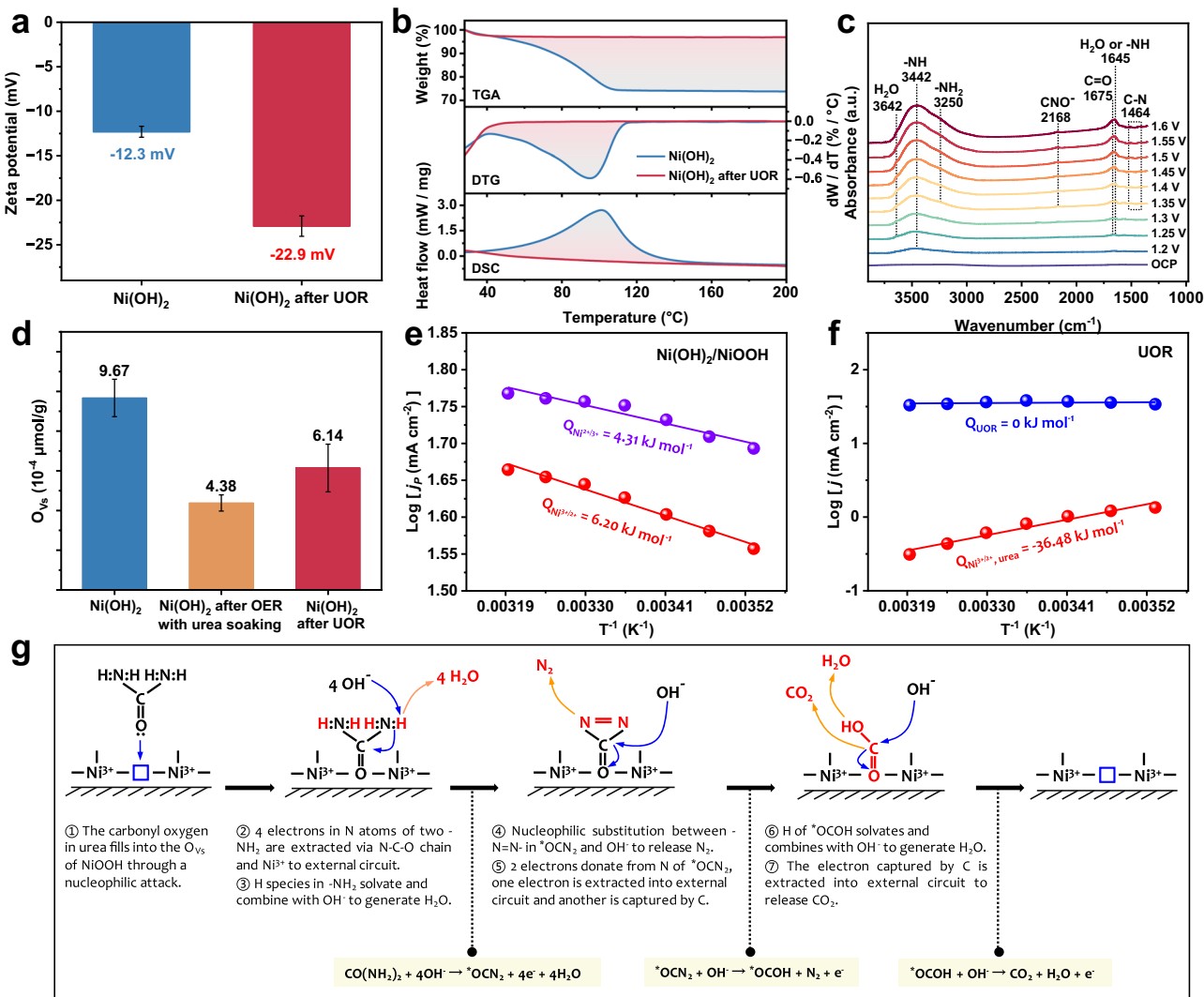

**Fig. 5 | The catalytic mechanism of UOR on Ni³⁺. a** Zeta potentials for Ni(OH)$_2$ before and after UOR. **b** Thermogravimetry analysis (TGA), derivative thermogravimetry analysis (DTG), and differential scanning calorimetry analysis (DSC) for Ni(OH)$_2$ before and after UOR. **c** In situ electrochemical ATR-SEIRAS on Ni(OH)$_2$ when anode potentials varied from 1.2 to 1.6 V with a rising step of 0.05 V in 1 M KOH + 0.33 M urea. A background spectrum was collected at open circuit potential (V$_{OCP}$). **d** Concentration of oxygen vacancies derived by quantitative EPR analysis for the as-prepared Ni(OH)$_2$ electrode, the Ni(OH)$_2$ electrode after UOR, and the Ni(OH)$_2$ electrode after OER with urea soaking. **e** The activation energy for electrooxidation of Ni(OH)$_2$ to NiOOH (Q$_{Ni^{2+/3+}}$) and electroreduction of NiOOH to Ni(OH)$_2$ (Q$_{Ni^{3+/2+}}$). **f** The activation energy for UOR (Q$_{UOR}$) and reduction of NiOOH by urea to Ni(OH)$_2$ (Q$_{Ni^{3+/2+},\ urea}$). **g** The UOR mechanism on Ni³⁺. The nucleophilic carbonyl oxygen fills into the coordinately unsaturated Ni site (O$_{Vs}$) in NiOOH via a nucleophilic attack process, thus building the isoenergetic channels to extract the electrons in -NH$_2$ of urea via -C-N-O- transfer chain to external circuit.

occur on Ni³⁺ species. No liquid products in ¹³C NMR spectra were detected after UOR, suggesting the high product selectivity for overall urea oxidation to CO$_2$, N$_2$, and H$_2$O (Supplementary Fig. 14b). The interactions between urea molecule and Ni(OH)$_2$ electrode were investigated by monitoring Zeta potentials due to that both the amino and carbonyl in urea are negatively charged, thus decreasing the Zeta potential if there is a preferential adsorption for urea than water to occur. The Ni(OH)$_2$ electrode prepared in aqueous solution would exhibit the water adsorbed surface and after UOR this electrode in the presence of water and urea would present the possible competition adsorption. Accordingly, the Zeta potentials of as-prepared Ni(OH)$_2$ electrode before and after UOR are able to indicate the adsorption of water and urea, respectively. After UOR, the Zeta potential significantly decreased from −12.3 to −22.9 mV (Fig. 5a), indicative of the strong interactions between Ni(OH)$_2$ and urea molecule with negatively charged groups. The preferential adsorption for urea than water on Ni(OH)$_2$ was also verified by thermogravimetry analysis (TGA). The

water adsorption on Ni(OH)$_2$ before and after UOR is about 28 wt. % and 3 wt. % (Fig. 5b), respectively, undoubtedly witnessing the preferential adsorption of urea. The inefficient water adsorption would originate from the frustrated electron exchange due to big energy diffidence between HOMO level of water and Fermi level of NiOOH.

The in situ electrochemical attenuated total reflection-surface enhanced infrared absorption spectroscopy (ATR-SEIRAS) was carried out to check the species evolution during UOR. As shown in Fig. 5c, deprotonation of amino group of urea first occurs to form -NH at 3442 cm⁻¹ and 1645 cm⁻¹ as the potential increases[12,35]. When potentials varied from 1.35 to 1.6 V, the vibration for CNO⁻ at 2168 cm⁻¹ appeared and gradually increased, indicating the UOR occurs once potential is above 1.3 V[35]. During UOR, gradually increasing -NH$_2$ at 3250 cm⁻¹, C = O at 1675 cm⁻¹, and H$_2$O at 3642 cm⁻¹ corresponded to adsorption enhancement of reactants, generation of intermediates, and accumulation of products, respectively[12,36]. And C-N at 1464 cm⁻¹ is slightly negative-going[12], indicating urea consumption. This result suggested

that oxidation of urea undergoes the removal of protons in -$NH_2$, probably the most thermodynamic favorite process[12,37]. Quantitative EPR analysis indicated that the $Ni(OH)_2$ electrode before and after UOR exhibited the similar concentration of oxygen vacancies ($O_{Vs}$, an EPR signal at g = 2.003)[38] to be $6.14-9.67 \times 10^{-4}$ μmol/$g_{catalyst}$ (Fig. 5d and Supplementary Fig. 15). However, soaking the as-prepared $Ni(OH)_2$ in urea will sharply decrease the $O_{Vs}$ concentration ($4.38 \times 10^{-4}$ μmol/$g_{catalyst}$), probably suggesting that the $O_{Vs}$ play a key factor for the interactions between urea and $Ni(OH)_2$ electrode. Indeed, the carbonyl oxygen in urea is the most nucleophilic atom. The $O_{Vs}$ in NiOOH electrode expose the coordinately unsaturated metal sites, as an electrophilic center, to directly interact with carbonyl oxygen of urea to trigger a nucleophile attack. Therefore, the FTIR results may suggest that the urea oxidation undergoes the deprotonation of the amino groups during the nucleophile attack to occur. Indeed, the UOR potential ($V_{10\ mA}$) on $Ni(OH)_2$ electrode is a linear function of pH with a negative slope of -54 mV $pH^{-1}$ (Supplementary Fig. 16), much close to -59 mV $pH^{-1}$ slope of theoretical $V_{10\ mA}$ versus pH for UOR under 25 °C and 1 atm, revealing the completely solvated protons during UOR. The activation energy (Fig. 5e and Supplementary Fig. 17), for electrooxidation of $Ni(OH)_2$ to NiOOH via $Ni(OH)_2 \rightarrow NiOOH + H^+ + e^-$, is 4.31 kJ $mol^{-1}$ ($Q_{Ni2+/3+}$), which is slightly lower than the 6.20 kJ $mol^{-1}$ ($Q_{Ni3+/2+}$) for the electroreduction of NiOOH to $Ni(OH)_2$, indicating that the PCET process possesses different kinetics for the $Ni(OH)_2$/NiOOH interconversion. However, the activation energy of UOR is 0 kJ $mol^{-1}$ ($Q_{UOR}$, Fig. 5f), clearly confirming that the $Ni^{3+}$ oxidizing urea is an isoenergetic electron transfer through electron tunneling between the nucleophilic atom in urea molecule and the active center of NiOOH (Inset in Fig. 1b). In the presence of urea, the activation energy for electroreduction of NiOOH to $Ni(OH)_2$ is -36.48 kJ $mol^{-1}$ ($Q_{Ni3+/2+,\ urea}$), meaning that the reduction of NiOOH by urea to $Ni(OH)_2$ is a spontaneous reaction process. The UOR reaction energetics also point to a standpoint that the UOR on $Ni^{3+}$ is a nonredox electron transfer process. Herein, we can conclude the mechanism of urea oxidation by $Ni^{3+}$ (Fig. 5g). When oxidizing $Ni(OH)_2$ to NiOOH, the nucleophilic carbonyl oxygen fills into the $O_{Vs}$ in NiOOH via a nucleophilic attack process, thus building the isoenergetic channels to transfer electrons. The 4 electrons in N atoms of two -$NH_2$ of urea are extracted via -C-N-O- transfer chain to external circuit through unoccupied $e_g$ orbitals of $Ni^{3+}$, while the 4 protons of two -$NH_2$ solvate and combine with $OH^-$ to generate $H_2O$ via a reaction of $CO(NH_2)_2 + 4OH^- \rightarrow {}^*OCN_2 + 4e^- + 4H_2O$[12]. Subsequently, nucleophilic substitution between -N=N- in ${}^*OCN_2$ and $OH^-$ occurs to release $N_2$ via a reaction of ${}^*OCN_2 + OH^- \rightarrow {}^*OCOH + N_2 + e^-$[39]. During N-N coupling to generate $N_2$, two electrons are donated from two N atoms of ${}^*OCN_2$. One electron is extracted into external circuit and the another is captured by C of ${}^*OCOH$. Finally, the H of ${}^*OCOH$ solvates and combines with $OH^-$ to generate $H_2O$, and the electron captured by C is extracted into external circuit to release $CO_2$ via a reaction of ${}^*OCOH + OH^- \rightarrow CO_2 + H_2O + e^-$[39].

## Universality of nonredox trivalent nickel catalysis

Identifying kinetically fast organic electrooxidation on nonredox $Ni^{3+}$-based catalysts is of great importance for the development of efficient energy conversions. The nucleophiles are commonly negatively charged or have at least one lone electron pair they can easily share to make a new chemical bond. The nitrogen or oxygen atoms in organic functional groups usually are negatively charged species. Accordingly, we check the availability of $Ni^{3+}$ oxidizing the organics with nucleophilic functional groups, such as hydroxyls (methanol, ethanol, and benzyl alcohol), carbonyls (formamide, urea, formaldehyde, glucose, formic acid, and N-acetyl glucosamine), and aminos (benzylamine and ammonia). The LSV polarization curves show that the $Ni^{3+}$ is highly active for these selected organics except the ammonia and formic acid (Supplementary Figs. 18, 19). DFT calculations have shown that combination of HOMO level of organics and $\Delta s_k$ values of nucleophilic

atoms is able to judge which organics are efficiently catalyzed by $Ni^{3+}$. As shown in Fig. 6 and Supplementary Fig. 20, the theoretical calculations indicated that the $Ni^{3+}$ can oxidize organics possessing the HOMO energy levels ranging from -7.4 to -6 eV (vs. Vacuum level) and the $\Delta s_k$ values of nucleophilic atoms ranging from -0.65 to -0.15. The calculated $\Delta s_k$ values revealed that the most nucleophilic atoms in these organics are the oxygen or nitrogen in the nucleophilic functional groups, such as hydroxyls, carbonyls, and aminos. The nucleophilic atoms take part in the formation of HOMO orbitals, meaning that the nucleophilic attack occurs via overlap of the nucleophile's HOMO with the 3d orbitals of $Ni^{3+}$ in NiOOH.

The combination (HOMO, $\Delta s_k$) for ammonium hydroxide ($NH_4OH$, a stable species for ammonia gas in water) and formic acid is (-4.82 eV, -0.932) and (-7.99 eV, -0.300), respectively. This means that the inefficient electrooxidation for ammonia and formic acid on $Ni^{3+}$ species is attributed to the more negative $\Delta s_k$ value of -0.932 and the deeper HOMO energy level of -7.99 eV, respectively. The HOMO level of ammonia is higher than the Fermi level of NiOOH, which allows the electron transfer to trigger a nucleophilic attack. Therefore, the inefficient $Ni^{3+}$ oxidizing ammonia would be attributed to that the too-strong nucleophilicity of the N atom in $NH_3$ tends to form a stable covalent bond with $H_2O$ in the form of $NH_4OH$. The oxygen atom in $NH_4OH$ is the most nucleophilic site and exhibits the strong nucleophilic attack to the electrophilic $Ni^{3+}$ center, thus limiting the reaction kinetics. However, the inefficient electrooxidation of formic acid results from its deep HOMO level far away from the Fermi level of NiOOH, inducing the high electron transfer barriers for molecule adsorption and conversion. The combination (HOMO, $\Delta s_k$) for $CO_2$ (-10.13 eV, -0.143) and $N_2$ (-11.55 eV, 0) does not meet the criterion of $Ni^{3+}$ oxidizing organics, well explaining why they are the final products for electrooxidation of most organics.

To evaluate the practical application potential of $Ni^{3+}$ oxidizing organics, the oxidation ability of $Ni^{3+}$ was further estimated. After polarizing the $Ni(OH)_2$ to NiOOH, introducing the organics will induce the rapid $V_{OCP}$ decay to a quasi-equilibrium potential of 1.26 V (Supplementary Fig. 21), a potential region for $Ni^{2+}$ generation, clearly indicating that the $Ni^{3+}$ oxidizing organics is a kinetically rapid spontaneous chemical reaction. This characteristic is useful for creating a decoupled hydrogen generation from biomass resources or organics-containing industrial wastewater. A possible energy-effective device is expected to be comprised of an anode to electrochemically oxidize $Ni(OH)_2$ to NiOOH and a cathode to reduce $H^+$ to $H_2$ in 1 M KOH electrolyte. The NiOOH anode can be reduced to $Ni(OH)_2$ through moving the electrode to organics-containing electrolyte to trigger a spontaneous chemical reaction between $Ni^{3+}$ and organics. Accordingly, we can divide the process into two steps: an electrochemical step that reduces water at the cathode to produce hydrogen and oxidizes the anode to form NiOOH in 1 M KOH electrolyte, followed by a spontaneous chemical step reduces the anode back to its initial state by oxidizing organics. The spatially separated two-step processes will achieve hydrogen production and oxidation of organics in different reaction chambers, thus benefiting to produce the high-purity products. The low potential requirement for $Ni(OH)_2$/NiOOH oxidation, usually below 1.4 V vs. RHE, is an inherently low energy-consuming process. And the spontaneous reaction of $Ni^{3+}$ oxidizing organics is kinetically rapid at room temperature, suggesting that this technique to couple with hydrogen production is low-cost and time-effective.

To summarize, we have elucidated the efficient nucleophilic electrooxidation mechanism on trivalent nickel, which is an extremely fast nonredox electron transfer center for accelerating nucleophilic attack from nucleophilic oxygen or nitrogen atoms in organics. Combination of HOMO levels and $\Delta s_k$ values of nucleophilic atoms can identify which organics that are capable of being oxidized by $Ni^{3+}$. The kinetically rapid organic electrooxidation may be achieved by the organics with a combination (HOMO, $\Delta s_k$) composed of HOMO levels

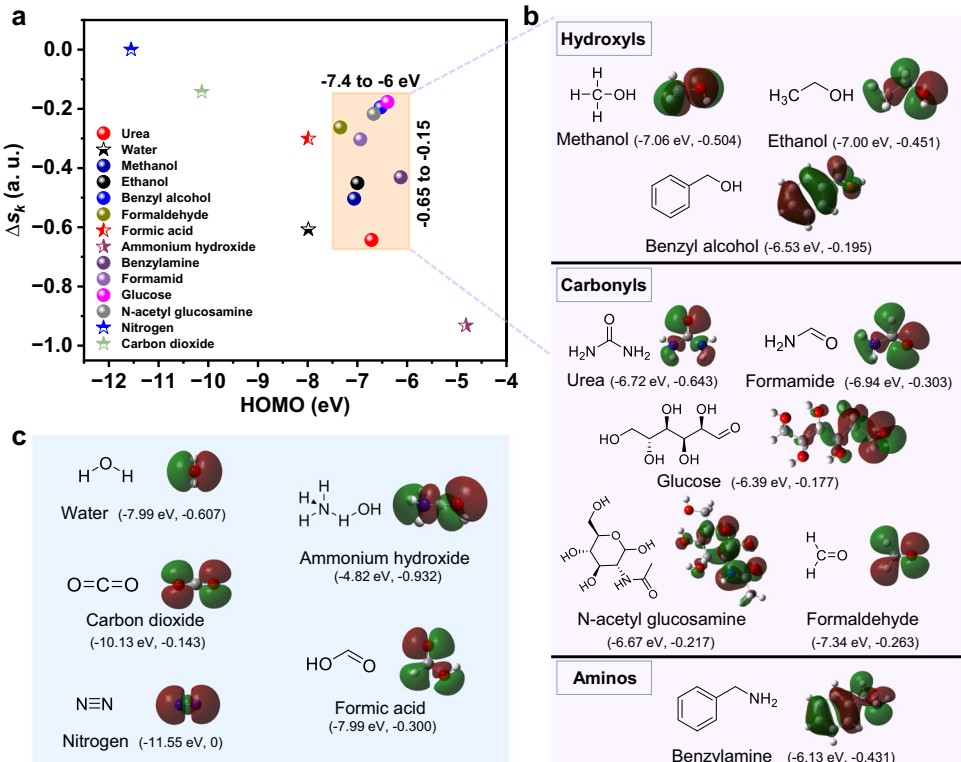

**Fig. 6 | The universality for electrooxidation of organics by Ni³⁺.** **a** The $\Delta s_k$ values of nucleophilic atoms and HOMO orbitals of molecules as the descriptors to screen the possible organics that are able to be efficiently oxidized by Ni³⁺. **b** The calculated HOMO orbitals and structural formula for organics that can be oxidized by Ni³⁺. **c** The calculated HOMO orbitals and structural formula for organics that can not be oxidized by Ni³⁺.

from −7.4 to −6 eV (vs. Vacuum level) and $\Delta s_k$ values of nucleophilic atoms ranging from −0.65 to −0.15. Our findings provide new insights into the catalysis of nickel-based catalysts useful in the development of efficient energy conversions.

## Methods
### Catalyst preparation
Potassium hydroxide (KOH) was purchased from Shanghai Aladdin Bio-Chem Technology Co., Ltd. (Shanghai, China). Nickel acetate tetrahydrate (Ni(CH₃COO)₂·4H₂O) was obtained from Shanghai Bidepharm Medical Technology Co., Ltd. (Shanghai, China). Methanol, ethanol, benzyl alcohol, formaldehyde, formic acid, formamide, ammonia, benzylamine, urea, glucose, N-acetyl glucosamine, dimethylglyoxime disodium salt octahydrate (C₄H₆N₂Na₂O₂·8H₂O) were purchased from Shanghai Aladdin Bio-Chem Technology Co., Ltd. (Shanghai, China). All the chemicals were analytical grade and used as received without further purification. Purified water was purchased from China Resources Cestbon Beverage Co., Ltd. (Shenzhen, China). Ni foam was purchased from Hefei Kejing Materials Technology Co., Ltd. (Hefei, China). Before depositing the Ni(OH)₂, a piece of Ni foam with a size of 2 × 1 cm² was in turn cleaned by 3.0 M hydrochloric acid, ultrapure water, and ethanol with sonication for 30 min and then dried in N₂ flow for subsequent use. Ni(OH)₂ electrode were synthesized by a one-step electrodeposition method. The samples were electrochemically deposited on a 1 × 1 cm² effective area of 2 × 1 cm² Ni foam. The nondeposited area was masked by insulated rubber tape to screen the electrolyte. Electrodeposition was carried out in a three-electrode system with a KCl-saturated Ag/AgCl electrode as the reference electrode and a platinum sheet as the counter electrode. The Ni(OH)₂ electrode was deposited in 50 mL electrolyte containing 0.025 M Ni(CH₃COO)₂·4H₂O. All electrochemical deposition experiments were carried out at a constant potential of −3.5 V vs. reversible hydrogen electrode (RHE) for 120 s. All the as-deposited electrodes were washed with deionized water and then purged with N₂.

### Material characterizations
The crystallographic structures of the as-prepared electrodes were checked by X-ray diffraction spectroscopy (XRD, Ultima III, Rigaku Corp., Japan) with Cu Kα radiation at 40 kV and 40 mA at a scan rate of 1° min⁻¹. Morphologies and microstructures were observed with scanning electron microscopy (FE-SEM, FEI NOVA NanoSEM230, USA) and transmission electron microscopy (TEM, JEOL 3010, Japan). The surface elemental compositions and chemical states were examined by X-ray photoelectron spectroscopy (XPS, ULVAC-PHI, PHI5000 Versa Probe, Japan) with monochromatized Al Kα excitation. The binding energy was corrected by referencing to the C 1 s spectrum at 284.6 eV, and a Shirley background was used for peak fitting. Ex situ Raman spectra were examined by Horiba T64000@514 nm (HORIBA Scientific, France) with an argon ion as laser light source. Ex situ Fourier transform infrared spectroscopy (FTIR) was performed on a Nicolet Nexus 870 infrared spectrometer. Nuclear magnetic resonance (NMR, Ascend 600 MHz, Bruker, Germany) was used to analyze the liquid products of the UOR, and 500 μL of electrolyte and 100 μL of D₂O were added in an NMR tube and mixed thoroughly before test. The 1D NMR signal was obtained at 298.6 K with 1024 numbers of scan, 0.9175 s acquisition time, and 2 s relaxation delay. The oxygen vacancies of the products were detected by electron paramagnetic resonance spectrometry (EPR, EMXplus X-band, Bruker, Germany). The thermogravimetric (TG), derivative thermogravimetry analysis (DTG), and differential scanning calorimetry (DSC) curves were examined by STA 449 F3 Jupiter (Netzsch, Germany). The selectivity and faradaic efficiency of the gaseous N₂ product of UOR in N₂-saturated electrolyte containing 1 M KOH + 0.33 M urea were analyzed by gas chromatography (GC, 8860, Agilent, USA). Zeta potential was detected by Zetasizer Nano ZS90 (Malvern, Worcestershire, UK). In situ UV-vis

absorption spectra were obtained with an ultraviolet-visible spectro-photometer (UV3600, Shimadzu, Japan) in a homemade three-electrode cuvette under the diffuse reflectance model using an integrating sphere.

## Electrochemical measurements

The electrochemical measurements on the as-prepared electrodes were performed in a three-electrode single cell containing 1 M KOH with/without organic molecule electrolyte with 90% iR compensation. The reference electrode was a Hg/HgO electrode filled with 1 M KOH solution, and a high-purity graphite rod was used as the counter electrode. Linear sweep voltammetry (LSV), cyclic voltammetry (CV), electrochemical impedance spectrum (EIS), and chronoamperometry were performed on a CHI 660 electrochemical workstation (Shanghai Chenhua Science Technology Corp., Ltd., Shanghai, China). LSV was performed at a scan rate of 5 mV s$^{-1}$ from 0.1 to 1 V vs. Hg/HgO. In situ EIS was recorded with a frequency range from 100 kHz to 0.01 Hz on potential windows for UOR from 1.1 to 1.6 V vs. RHE with an increment of 50 mV. The open-circuit potential ($V_{OCP}$) decay was tested immediately after electroxidation at 70 mA cm$^{-2}$ on a CHI 730 electrochemical workstation (Shanghai Chenhua Science Technology Corp., Ltd., Shanghai, China). All measured potentials vs. Hg/HgO were converted to reversible hydrogen electrode (RHE) by employing the Nernst function:

$$E_{RHE} = E_{Hg/HgO} + 0.095 + 0.0592 \times pH \qquad (1)$$

## In situ XANES

The in situ X-ray absorption near edge structure (XANES) spectra at the Ni K-edge were collected on TableXAFS-500A (Anhui Specreation Instrument Technology Co., Ltd., Hefei, China). Monochromatized X-ray beam was provided by an X-ray tube and a spherically bent crystal assembled on the R250 mm Rowland circle. The Bragg angle was about 78° with the Si(551) lattice plane. All the spectra were recorded in transmission mode with a photon energy resolution of about 1 eV. The photon energies were calibrated to the first inflection point of the K edge from Ni foil at 8333 eV. The sample for in situ measurement was Ni(OH)$_2$ on carbon cloth (CC) which was electrochemically deposited at a constant potential of −3.5 V for 600 s in 100 mL electrolyte containing 0.025 M Ni(CH$_3$COO)$_2$•4H$_2$O. The in situ XANES spectra were acquired by amperometric measurements at open circuit potential ($V_{OCP}$) and potential windows from 1.3 to 1.5 V vs. RHE in 1 M KOH + 0.33 M urea, which were conducted on a CHI660E electrochemical workstation (Shanghai Chenhua Instrument Co.). The acquired XANES data were normalized using the ATHENA module implemented of Demeter software packages.

## In situ Raman measurements

In situ Raman measurements were performed using XPlora PLUS Raman spectrometer (HORIBA Scientific, France), and the excitation wavelength of the laser was 532 nm. Ni(OH)$_2$ electrode was assembled into a homemade spectroelectrochemical cell as the working electrode, Hg/HgO was used as reference electrode, and high-purity graphite served as the counter electrode. The 1 M KOH + 0.33 M urea electrolyte was saturated with high-purity Ar before testing. The in situ Raman spectra were acquired by amperometric measurements at open circuit potential ($V_{OCP}$) and potential windows from 1.15 to 1.5 V vs. RHE, which were conducted on a CHI660E electrochemical workstation (Shanghai Chenhua Instrument Co.). Each spectrum was obtained with an integration time of 60 s and twice accumulations.

## In situ ATR-SEIRAS measurements

The surface-enhanced infrared absorption spectroscopy (SEIRAS) with the attenuated total reflection (ATR) configuration was employed to detect the intermediates of UOR. The electrochemical ATR-SEIRAS measurements were performed on a Thermo Nicolet 8700 spectrometer equipped with mercury-cadmium-telluride detector cooled by liquid nitrogen. Chemical deposition of Au thin film ( ~ 60 nm) on the Si prism was prepared by a two-step wet chemical process. Before chemical deposition of Au, the Si prism surface for IR reflection was polished with diamond suspension and cleaned in deionized water with sonication. Then the prism was soaked in a piranha solution (7:3 volumetric ratio of 98 % H$_2$SO$_4$ and 30 % H$_2$O$_2$) for 2 h. The catalyst ink was prepared by dispersing 5 mg Ni(OH)$_2$ into a mixing solution of 975 μL H$_2$O and isopropyl alcohol (volume ratio = 1), and 25 μL Nafion solution (5 wt %) in an ultrasonic bath for 30 min. 30 μL ink was deposited and dried on the Au-film working electrode, then the ink-coated prism was assembled into a homemade spectro-electrochemical cell as the working electrode, Hg/HgO was used as reference electrode, which was introduced near the working electrode via a Luggin capillary, a Pt mesh (1 × 1 cm$^2$) was serving as the counter electrode. The electrolyte was Ar-saturated 1 M KOH + 0.33 M urea. The electrolyte was circulated by peristaltic pump to avoid gas accumulation affecting signal collection. The electrochemical ATR-SEIRAS was acquired with chronoamperometric tests, which were conducted using a CHI660E electrochemical workstation (Shanghai Chenhua Instrument Co.) at open circuit potential ($V_{OCP}$) and potential windows from 1.2 to 1.6 V vs. RHE. The spectral resolution was 4 cm$^{-1}$ for all the measurements. Each spectrum was measured by superimposing 200 interferograms. The spectra were acquired with the processing method that comes with the program by equation:

$$A = -\log(R_{Es}/R_{ER}) \qquad (2)$$

where A is absorbance, $R_{Es}$ is reflection under studied potentials, and $R_{ER}$ is reflection at open circuit potential. With this processing method, the positive-going spectra correspond to the generative or increasing species, and the negative-going spectra correspond to the consuming or decreasing species.

## In situ UV-vis absorption measurements

In situ UV-vis absorption spectra were obtained with an ultraviolet-visible spectrophotometer (UV3600, Shimadzu, Japan) in a homemade three-electrode cuvette under the diffuse reflectance model using an integrating sphere. The reference electrode was a KCl-saturated Ag/AgCl electrode with a diameter of 4 mm, and a high-purity Pt wire was used as the counter electrode. The chronoamperometry operation was performed on a CHI660E electrochemical workstation (Shanghai Chenhua Instrument Co.). All measured potentials vs. Ag/AgCl were converted to reversible hydrogen electrode (RHE) by employing the Nernst function:

$$E_{RHE} = E_{Ag/AgCl} + 0.197 + 0.0592 \times pH \qquad (3)$$

## The fs-TAS measurements

The Ni(OH)$_2$ with and without adsorption of urea were coated onto 2 × 2 cm$^2$ high-transmittance optical quartz slides by spin-coating method. The femtosecond transient absorption spectra (fs-TAS) were collected using a commercial fs-TAS spectrometer (Helios, Ultrafast System LLC, USA). All experiments were performed at room temperature. The 800 nm fundamental beam was generated from a Ti: sapphire laser amplifier system (<100 fs, > 7 mJ/pulse, and 1 kHz, Astrella, Coherent Inc., USA). The 320 nm pumping pulses, generated from an optical parametric amplifier (OPA-Solo, Coherent Inc., USA), were chopped at 500 Hz and used as the reference frequency for the lock-in amplifier. Less than 10% of the fundamental 800 nm laser pulses were attenuated with a neutral density filter and focused into a 2 mm thick sapphire window to generate a white light continuum

(WLC) from 320 to 750 nm used for probe beam. The probe beam was focused with an Al parabolic reflector onto the sample. The time delay between the pump and probe pulses was controlled by a motorized delay stage. Data are analyzed with the Surface Xplorer software.

### DFT calculations

The density functional theory (DFT) calculations were performed with the DMol³ software package[40,41]. The generalized gradient approximation (GGA) of PBE functional was used for describing the exchange-correlation energy[42]. The double numerical plus d-function (DND) basis set was used to optimize all spin-unrestricted structures and the Ni ($3d^5 4s^2$) electrons were treated as valence electrons using the effective core potential (ECP) method[43,44]. Additionally, the DFT-D2 method proposed by Grimme was adopted to describe the van der Waals interactions[45,46]. NiOOH has a triclinic structure with space group P1[47]. The calculated lattice constant is $a = 4.951$, $b = 13.477$, and $c = 25.000$ Å, respectively. The NiOOH (100) surface with coordinatively unsaturated Ni sites, a highly active surface, was selected for DFT calculations. The created (001) surface slab contains three atomic layers composed of 12 Ni atoms, 24 O atoms, and 6 H atoms. Considering the important effects of oxygen vacancies on the electrical and catalytic properties of NiOOH[20,21], the surface slab for NiOOH (100) facet with oxygen vacancies (denoted as NiOOH (100)·$O_v$) was also built by removing one O atom on the NiOOH (100) facet (Supplementary Fig. 3a). The structure optimization was based on an energy tolerance of $1 \times 10^{-5}$ Ha (1 Ha ≈ 27.2114 eV), a force tolerance of 0.002 Ha Å⁻¹ and a displacement tolerance of 0.005 Å. The quantum chemistry calculations of the organic molecules including complete geometrical optimization were performed using B3LYP hybrid density functional with 6-31 G or 6-31 G ($d$, $p$) basis set of Gaussian 09 W. Particularly, 6-31 G ($d$, $p$) basis set was used for the ring compounds, benzyl alcohol, benzylamine, and N-acetyl glucosamine, to enhance the spatial softness of valence orbitals.

To determine the possible nucleophilic sites in organics and the thermodynamic potentials for electrooxidation of organics, we calculated the dual local softness ($\Delta s_k$) and the highest occupied molecular orbital (HOMO) for all selected organics[48].

The condensed Fukui function for $k$ atom is calculated by[49]

$$f_k^+ = q_{N+1}^k - q_N^k \text{ (for a nucleophile attack)} \qquad (4)$$

$$f_k^- = q_N^k - q_{N-1}^k \text{ (for an electrophile attack)} \qquad (5)$$

where, $q^k$ is atomic charge, $N$ is the number of electrons[48,50].

Condensed dual descriptor (CDD) can be calculated by[17]

$$\triangle f_k = f_k^+ - f_k^- = q_{N+1}^k - 2q_N^k + q_{N-1}^k \qquad (6)$$

The dual local softness for $k$ atom ($\Delta s_k$), representing both intramolecular and intermolecular nucleophilicity of $k$ atom[17], was calculated as the condensed dual descriptor of $\Delta f_k$ multiplied by the molecular softness ($S$)

$$\triangle s_k = S \triangle f_k \qquad (7)$$

### Data availability

The data used in this study are available in the figshare database under [https://doi.org/10.6084/m9.figshare.24501238].

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

## Acknowledgements
This work was supported primarily by the Scientific and Technological Innovation Project of Carbon Emission Peak and Carbon Neutrality of Jiangsu Province (No. BE2022028-1), the National Natural Science Foundation of China (Grant Nos. 52272217, 22372078, 52073006, 51872135, 51572121, and 21633004), the Innovative Science and Technology Platform Project of Cooperation between Yangzhou City and Yangzhou University (No. YZ2020263).

## Author contributions
The work was conceived and designed by S.Y. and Y.Y.; Y.Y., R.W., Q.Z., and J.Z. carried out the experiments; Y.Y. performed the DFT calculations; W.H. performed the XAS measurements; Z.Z. discussed the experimental ideas; S.Y. and Y.Y. drafted the manuscript and revised the manuscript; all authors discussed the results.

## Competing interests
The authors declare no competing interests.
