## [Peer Review File · Nature Communications]

REVIEWER COMMENTS

Reviewer #1 (Remarks to the Author):

Yan et al report nonredox trivalent nickel catalyzing nucleophilic electrooxidation where clearly show that the electrooxidation of organics on Ni³⁺ species does not follow the Ni^{2+/3+} redox-mediated electron transfer mechanism. Both theoretical and experimental techniques were characterized to verify above viewpoint. This work is very interesting for the researchers studying the nickel electrocatalysis, and the discovery seems solid and credible. Therefore, I recommend this work published in Nature Communications after minor revisions below:

1. A scheme for this work, including materials, theoretical and experimental techniques, must be represented in a one scheme for emphasizing the highlight point, thought and whole process for texting, in order to improve the readability.
2. In Figure 3f, it seems that the electron transfer mechanism in Ni-O-Ni configuration like the double-exchange/super-exchange effect in Mn-O-Mn system, what is the differences between them?
3. It should be added DTG curves in Figure 4b, and corresponding DSC test should be added to further verify the opinion.
4. There is some inconsistency in Electrochemical details. For example, in "electrochemical measurement" part, CHI 660 and Hg/HgO as reference; Meanwhile, in "In situ UV-vis absorption measurements" part, CHI660E and Ag/AgCl as reference. Why aren't their testing conditions as consistent as possible?

Reviewer #2 (Remarks to the Author):

The work by Yan, Zou and co-workers aims at mapping the water and organics oxidation reactions catalyzed by trivalent Ni-based catalysts, and addressing the differences in their mechanisms that are currently under heavy debate. An array of characterization techniques are employed to map the oxygen evolution reactions and urea oxidation reactions on Ni(OH)₂ electrodes. Proof of the UOR reactions driven by Ni³⁺ electron transfer kinetics for the Ni³⁺/urea oxidation reactions are further provided. The manuscript is very well written, and the experiments have been thoroughly conducted. I recommend publication of this work in Nature Communications after the following comments are addressed:

1. I would recommend the authors to corroborate the existence of Ni³⁺ species during UOR with X-ray absorption measurements as well if possible.
2. The EPR data illustrated in Figure S12 are too weak to obtain reliable quantitative analysis of the Ni(OH)₂ electrode before and after UOR. The error bars have further not been provided. This data set needs to be repeated before the manuscript can be publishable.
3. The following sentences are difficult to read and not well constructed. Please correct:
 - a. "To circulate the electrochemical oxidation and chemical regeneration of Ni(OH)₂ will achieve a decoupled hydrogen production."
 - b. "And the room temperature chemical reaction between Ni³⁺ and organics is a kinetically rapid...."

Reviewer #3 (Remarks to the Author):

This manuscript describes the electrooxidation of organic compounds by Ni(OH)₂. It is found that in contrast to water oxidation, the reaction occurs already at the NiOOH state. A clear correlation between the HOMO energy of the substrate and/or the softness of the coordinating atom of the substrate with the tendency to get oxidized at the Ni(III) state. The findings are very interesting and it seems clear that the active state is NiOOH. The correlation with the molecular descriptors gives easy predictability and should be of interest to a broader community. My assessment of the computations is that they were performed at a satisfactory level. They are quite simple but contain the necessary information to predict the reactivity. One small remark is that the calculations were performed at different levels of theory for the small organics and the solid, and perhaps some care should be taken when comparing energy levels. However, I believe that the conclusion would be identical if both systems were calculated at the same level, except that the numbers would shift slightly. Overall I believe that this is a very interesting manuscript with well founded conclusions in a field that is really emerging at the moment. I recommend that it is published as is.

Reviewer #4 (Remarks to the Author):

Yan S. and coll. investigate in this paper the mechanism of electrocatalytic urea oxidation at Ni(OH)₂ electrodes. In their study, the authors support that this mechanism shuttles via non-redox processes

leaving the oxidation state of Ni(+III) untouched during catalysis. The study combines computational calculations to assess the softness and HOMO energy levels of the substrate and surface sites, electrochemical measurements of i/E curves, open-circuit potentials, coupling to Raman spectroscopy, and impedance spectroscopy.

The quality of the data is of overall good level and the presentation of this data is well done. However, I find that the hypothesis brought by the authors, that UOR proceeds through Ni centers that remain in their +III oxidation state, is not sufficiently strongly supported.

I develop here my points:

- First, discussing catalysis requires that a catalytic outcome is actually probed. This report does not contain results of electrocatalytic outcome (product quantification, selectivity, faradaic efficiency) for UOR with the electrodes used by the authors, nor an obvious reference to the same system. Without that, discussing mechanistic considerations seems vain.
- OCP and Raman experiments (Fig. 2d, e) show that Ni(III) species can actually oxidize urea.
- It is very obscure to me why the authors resorted to cell voltage measurements to estimate kinetics at the anode. This procedure is more likely to introduce kinetics at the cathode in the whole problem. Similar steady-state OCP values for voltages vs ref and vs CE does not mean that the transient kinetics at the cathode can be ignored. I thus do not see how the authors conclude that the cell voltage depends on Ni²⁺/Ni³⁺ kinetics. In addition, it was already proposed above that this kinetics is slow, likely due to phase change (hysteresis in LSV).
- The experiments with chelation by a dmg salt are not conclusive. Under catalytic conditions, the potential applied is always more positive than $E(\text{Ni(III/II)})$ and thus formed Ni(II) species short-lived. When CV cycling, it is not the case, since potential excursions below $E(\text{Ni(III/II)})$ quantitatively convert the species into Ni(II) over extended time. At most, these comparative experiments allow to conclude that there is no long-lived soluble Ni(II) intermediate in UOR and are by far not a “solid evidence [] that urea oxidation does not undergo the Ni²⁺/Ni³⁺ redox couple”.

Some secondary issues:

- Fig 4d. IR: C-N and C-O vibrations are almost baseline. Also, is there rationale why these vibrations are observed but no N-H ones?
- Sup. Fig. 13: the UOR equation is not equilibrated (O). I would at this pH write it with H₂O, not H⁺. And thus the Nernst equation should also be corrected.
- An “onset” potential has no meaning. One should better report the potential at a given current density.

- The text is often very hard to read and understand. An important style-editing would be needed before publication.

In conclusion, the main claim of the authors does not appear to be backed enough by data and there is at this stage no solid experimental evidence that sustains the mechanistic hypothesis. In my opinion, the strongest claim that can currently be made is that Ni(II) are not long-lived intermediates (/resting states), which is however not surprising given the applied potentials. If the authors wish to to further address the implication of the Ni(III) and Ni(II) species in UOR, I would suggest performing additional experiments, such a with operando spectroscopies that enable addressing oxidations state (XAS, XPS,...). On these grounds, I have selected reject and resubmit.

Response to Reviewer's comments

(NCOMMS-23-34704A)

Reviewer #1 (Remarks to the Author):

Yan et al report nonredox trivalent nickel catalyzing nucleophilic electrooxidation where clearly show that the electrooxidation of organics on Ni^{3+} species does not follow the $\text{Ni}^{2+/3+}$ redox-mediated electron transfer mechanism. Both theoretical and experimental techniques were characterized to verify above viewpoint. This work is very interesting for the researchers studying the nickel electrocatalysis, and the discovery seems solid and credible. Therefore, I recommend this work published in Nature Communications after minor revisions below:

1. A scheme for this work, including materials, theoretical and experimental techniques, must be represented in a one scheme for emphasizing the highlight point, thought and whole process for texting, in order to improve the readability.

Reply: Thanks for your valuable suggestions for improving the readability of our study. A scheme of our work was provided as Fig. 1c in the main text.

Fig. 1 | The challenges in understanding UOR and OER on Ni(OH)₂ electrode. c, A scheme to describe our strategy to discover the electrooxidation mechanism of organics on Ni^{3+} active center with a nonredox electron transfer.

2. In Figure 3f, it seems that the electron transfer mechanism in Ni-O-Ni configuration like the double-exchange/super-exchange effect in Mn-O-Mn system, what is the differences between them?

Reply: Thanks for your deep thinking on the electron transfer mechanism. The XAS data clearly show that single Ni^{3+} species exist in a distorted NiO_6 octahedron of NiOOH (*J. Electrochem. Soc.*, 1996, 143, 1613-1616). And in our study, we have demonstrated by various methods including In situ XANES, in situ UV-vis absorption spectra, and in situ Raman spectra, that the UOR is catalyzed by Ni^{3+} in NiOOH. The double exchange mechanism, an exchange interaction between two metal ions with parallel alignment of the spins separated by an

oxygen ion, was first found in the $\text{Mn}^{3+}\text{-O-Mn}^{3+}$ (or Mn^{4+}) by Zener in 1951 (*Phys. Rev.* 1951, 82, 403-405). According to the ligand field theory, a Jahn-Teller distortion of octahedrally coordinated NiO_6 units in NiOOH suggests that the Ni species in NiOOH is low-spin $3d^7 \text{Ni}^{3+}$ with the $t_{2g}^6 e_g^1$ electronic configuration (*Adv. Funct. Mater.* 2022, 32, 2111234; *Adv. Mater.* 2023, 35, 2203420), thus following a double exchange mechanism, the same as the Mn-O-Mn.

3. It should be added DTG curves in Figure 4b, and corresponding DSC test should be added to further verify the opinion.

Reply: Thanks for your help. We have updated the data in **Figure 4b**. The TGA, DTG, and DSC curves were added to the **Fig. 4b**.

Fig. 4 | The catalytic mechanism of UOR on Ni^{3+} . **b**, Thermogravimetry analysis (TGA), derivative thermogravimetry analysis (DTG), and differential scanning calorimetry analysis (DSC) for $\text{Ni}(\text{OH})_2$ before and after UOR.

4. There is some inconsistency in Electrochemical details. For example, in “electrochemical measurement” part, ; Meanwhile, in “In situ UV-vis absorption measurements” part, CHI660E and Ag/AgCl as reference. Why aren't their testing conditions as consistent as possible?

Reply: Thanks for carefully reading our study. Indeed, in our study, most electrochemical measurements, including LSV, CV, and OCP, were carried out on CHI 660 and used Hg/HgO as reference. However, for the in situ UV-vis absorption measurement, the reference electrode used is limited by the size of the in situ reaction cell. A cuboid-shaped quartz cuvette, with interior dimensions of 10 mm (width) and 10 mm (length), was used as the in situ reaction cell. So, the Ag/AgCl with a diameter of 4 mm was able to be input into the cell as a reference electrode compared Hg/HgO electrode with a diameter of 6 mm. That is, the Ag/AgCl reference electrode used in this in situ measurement is to ensure that all the electrodes including reference electrode, working electrode, and counter electrode can be put into the small quartz cuvette.

Reviewer #2 (Remarks to the Author):

The work by Yan, Zou and co-workers aims at mapping the water and organics oxidation reactions catalyzed by trivalent Ni-based catalysts, and addressing the differences in their mechanisms that are currently under heavy debate. An array of characterization techniques are employed to map the oxygen evolution reactions and urea oxidation reactions on Ni(OH)₂ electrodes. Proof of the UOR reactions driven by Ni³⁺ electron transfer kinetics for the Ni³⁺/urea oxidation reactions are further provided. The manuscript is very well written, and the experiments have been thoroughly conducted. I recommend publication of this work in Nature Communications after the following comments are addressed:

1. I would recommend the authors to corroborate the existence of Ni³⁺ species during UOR with X-ray absorption measurements as well if possible.

Reply: Thanks for your help. The X-ray absorption near edge structure (XANES) spectra were carried out and provided in **Figure 3d**. The photon energy was calibrated by the first peak maximum of the first derivative of a nickel foil (8333.0 eV). The X-ray absorption near edge structure (XANES) spectra revealed that the intensity of the white line decreases with polarizing the Ni(OH)₂ electrode from 1.3 to 1.5 V in 1 M KOH + 0.33 M urea, suggesting the formation of higher-valence Ni species due to that the high-valence Ni inducing distortion in the NiO₆ octahedral configuration will result in a decrease in the white line intensity (*J. Electrochem. Soc.* 1990,137, 383-388). Meanwhile, the edge energy (measured at half height) for the electrode at 1.3 and 1.5 V is the same as those of the Ni(OH)₂ (8342.6 eV) and NiOOH (8344.5 eV), respectively. This edge position suggests a +3 nickel oxidation state for the electrode during UOR (*J. Am. Chem. Soc.* 2012, 134, 6801-6809). The in situ XANES results confirmed that during UOR the Ni³⁺ is UOR-active and is likely to keep a constant valence state, in good agreement with the results from in situ Raman and in situ UV-vis absorption.

Fig. 3 | Evidences to trivalent nickel as active species for urea electrooxidation. d, Normalized in situ Ni K-edge XANES spectra for polarizing Ni(OH)₂ electrode at different potentials in 1 M KOH + 0.33 M urea, referenced to Ni foil and NiOOH.

2. The EPR data illustrated in Figure S12 are too weak to obtain reliable quantitative analysis of

the Ni(OH)₂ electrode before and after UOR. The error bars have further not been provided. This data set needs to be repeated before the manuscript can be publishable.

Reply: Thanks for your help. To strengthen the EPR signals, the EPR data were updated on an advanced Bruker (EMXplus X-band, Germany) electron paramagnetic resonance spectrometer. The error bars were provided based on the statistical data from three samples.

Fig. 5 | The catalytic mechanism of UOR on Ni³⁺. d, Concentration of oxygen vacancies derived by quantitative EPR analysis for the as-prepared Ni(OH)₂ electrode, the Ni(OH)₂ electrode after UOR, and the Ni(OH)₂ electrode after OER with urea soaking.

Supplementary Figure 12 | EPR signals for the as-prepared Ni(OH)₂ electrode, the Ni(OH)₂ electrode after UOR, and the Ni(OH)₂ electrode after OER with urea soaking.

3. The following sentences are difficult to read and not well constructed. Please correct:
- “To circulate the electrochemical oxidation and chemical regeneration of Ni(OH)₂ will achieve a decoupled hydrogen production.”
 - “And the room temperature chemical reaction between Ni³⁺ and organics is a kinetically rapid....”

Reply: Thanks for your help. We have revised these sentences as follows:

- Accordingly, we can divide the process into two steps: an electrochemical step that reduces

water at the cathode to produce hydrogen and oxidizes the anode to form NiOOH in 1 M KOH electrolyte, followed by a spontaneous chemical step reduces the anode back to its initial state by oxidizing organics. The spatially separated two-step processes will achieve hydrogen production and oxidation of organics in different reaction chambers, thus benefiting to produce the high-purity products.

b. And the spontaneous reaction of Ni³⁺ oxidizing organics is kinetically rapid at room temperature, suggesting that this technique to couple with hydrogen production is low-cost and time-effective.

Reviewer #3 (Remarks to the Author):

This manuscript describes the electrooxidation of organic compounds by Ni(OH)₂. It is found that in contrast to water oxidation, the reaction occurs already at the NiOOH state. A clear correlation between the HOMO energy of the substrate and/or the softness of the coordinating atom of the substrate with the tendency to get oxidized at the Ni(III) state. The findings are very interesting and it seems clear that the active state is NiOOH. The correlation with the molecular descriptors gives easy predictability and should be of interest to a broader community. My assessment of the computations is that they were performed at a satisfactory level. They are quite simple but contain the necessary information to predict the reactivity. One small remark is that the calculations were performed at different levels of theory for the small organics and the solid, and perhaps some care should be taken when comparing energy levels. However, I believe that the conclusion would be identical if both systems were calculated at the same level, except that the numbers would shift slightly. Overall I believe that this is a very interesting manuscript with well founded conclusions in a field that is really emerging at the moment. I recommend that it is published as is.

Reply: Thank you very much for your affirmation and approval to our research. Indeed, as pointed out by the Reviewer, the DFT calculations for Fermi level of NiOOH and HOMO level of organics were performed by different methods on the DMol³ software package and Gaussian 09W, respectively. For comparing energy levels, all the energy levels were referenced to the vacuum level.

Reviewer #4 (Remarks to the Author):

Yan S. and coll. investigate in this paper the mechanism of electrocatalytic urea oxidation at Ni(OH)₂ electrodes. In their study, the authors support that this mechanism shuttles via non-redox processes leaving the oxidation state of Ni^(+III) untouched during catalysis. The study combines computational calculations to assess the softness and HOMO energy levels of the substrate and surface sites, electrochemical measurements of i/E curves, open-circuit potentials, coupling to Raman spectroscopy, and impedance spectroscopy.

The quality of the data is of overall good level and the presentation of this data is well done. However, I find that the hypothesis brought by the authors, that UOR proceeds through Ni

centers that remain in their +III oxidation state, is not sufficiently strongly supported.

I develop here my points:

- First, discussing catalysis requires that a catalytic outcome is actually probed. This report does not contain results of electrocatalytic outcome (product quantification, selectivity, faradaic efficiency) for UOR with the electrodes used by the authors, nor an obvious reference to the same system. Without that, discussing mechanistic considerations seems vain.

Reply: Thank you very much for your help. To confirm that the UOR on Ni(OH)₂ electrode is a catalytic reaction, we have checked the product quantification, selectivity, and faradaic efficiency. The electrooxidation of urea follows a reaction of $\text{CO}(\text{NH}_2)_2 + 6\text{OH}^- \rightarrow \text{N}_2 + \text{CO}_2 + 5\text{H}_2\text{O} + 6\text{e}^-$. This means that the UOR products are the gaseous N₂ and CO₂ and liquid H₂O. Considering that both the H₂O and CO₂ are soluble in 1 M KOH, the product, N₂, was analyzed by gas chromatography (GC) to confirm the selectivity and faradaic efficiency of UOR. The Faraday efficiency for N₂ is 94.5 % (near 100%, the deviation mainly resulted from the N₂ dissolving in electrolyte) at potentials above 1.4 V as shown in **Supplementary Fig. 12a**. The electron transfer amounts for N₂ generation were equal to the amount of electric charge that passed through the electrode during UOR, suggesting a UOR catalytic reaction to occur. Furthermore, the selectivity of UOR was confirmed by the ¹³C NMR spectra to detect the liquid products of long-time UOR. As shown in **Supplementary Fig. 12b**, no liquid products were detected after UOR, suggesting the high product selectivity for overall urea oxidation to CO₂, N₂, and H₂O.

Supplementary Figure 12 | a, The N₂ faradaic efficiency at anodic potentials between 1.35 and 1.5 V. **b**, The ¹³C NMR spectra to identify the UOR product.

- OCP and Raman experiments (Fig. 2d, e) show that Ni(III) species can actually oxidize urea.
- It is very obscure to me why the authors resorted to cell voltage measurements to estimate kinetics at the anode. This procedure is more likely to introduce kinetics at the cathode in the whole problem. Similar steady-state OCP values for voltages vs ref and vs CE does not mean that the transient kinetics at the cathode can be ignored. I thus do not see how the authors conclude that the cell voltage depends on Ni^{2+/3+} kinetics. In addition, it was already proposed

above that this kinetics is slow, likely due to phase change (hysteresis in LSV).

Reply: Thank you very much for your help. The LSV curve told us the occurrence of phase transition of $\text{Ni}(\text{OH})_2$ to NiOOH , which includes a coupled information of both thermodynamic and kinetic processes. In particular, during UOR, the kinetics for phase transition of $\text{Ni}(\text{OH})_2$ to NiOOH is completely coupled with the UOR kinetics. Therefore, we carried out the cell voltage measurements to further estimate kinetics at the anode. Indeed, as pointed out by the Reviewer, this procedure is possible to introduce kinetics at the cathode. Here, we used the Pt electrode as cathode to minimize the effects of cathodic kinetics on cell voltage. To clearly demonstrate the fast kinetics of high-activity Pt electrode, we use the Pt foil as both the anode and the cathode and monitor the cell voltage when periodically altering the anodic potentials between 1.6 and 1.65 V, a potential window for UOR occurring on Pt electrode. As shown in **Supplementary Figs. 7b and 7c**, no visible cell voltage decay can be observed when periodically altering the anodic potentials, confirming that the possible electrochemical processes on Pt are kinetically rapid. Accordingly, in our case of $\text{Ni}(\text{OH})_2$ anode and Pt cathode, the cell voltage decay is totally resulting from the sluggish kinetics of the anode.

Similarly, as for OCP measurement, the voltage decay of the anode was monitored in a three-electrode system, that is, a reference electrode, Hg/HgO , was used to monitor the voltage decay of the anode and the Pt foil was used as the cathode. In the three-electrode testing system, the electrochemical processes on the Pt electrode are kinetically rapid, as shown in **Supplementary Figs. 7b and 7c**. The transient kinetics at the Pt cathode is rapid and thus its effects on kinetics of the anode can be ignored.

Supplementary Figure 7 | a, Cell voltage decay of Pt_{anode}-Pt_{cathode} after periodically altering the anodic potentials between 1.6 and 1.65 V with a stay time of 10 s at every potential point. **b**, The time-dependent cell voltage equilibrium of Pt_{anode}-Pt_{cathode} when periodically altering the anodic potentials between 1.6 and 1.65 V.

Abruptly altering the anodic potentials between 1.6 and 1.65 V on the Pt foil, the cell voltage immediately reaches the equilibrium state, indicating fast transient kinetics on both cathode of Pt foil and anode of Pt foil, to rapidly adjust the Fermi level of the electrode, thus sensitively adapting to the potential changes. These facts confirmed that the cell voltage is mainly dominated by the anodic potentials.

- The experiments with chelation by a DMG salt are not conclusive. Under catalytic conditions, the potential applied is always more positive than $E(\text{Ni(III/II)})$ and thus formed Ni(II) species short-lived. When CV cycling, it is not the case, since potential excursions below $E(\text{Ni(III/II)})$ quantitatively convert the species into Ni(II) over extended time. At most, these comparative experiments allow to conclude that there is no long-lived soluble Ni(II) intermediate in UOR and are by far not a “solid evidence [] that urea oxidation does not undergo the $\text{Ni}^{2+}/\text{Ni}^{3+}$ redox couple”.

Reply: Thank you very much for your help. In our study, the experimental evidence chains were focused on ruling out a possibility of electron transfer during UOR via $\text{Ni(OH)}_2/\text{NiOOH}$, a phase transformation process. So, in order to avoid the possible confusing, we revised the “ $\text{Ni}^{2+}/\text{Ni}^{3+}$ redox couple” as “ $\text{Ni(OH)}_2/\text{NiOOH}$ redox couple” in whole main text.

In addition, the main worry from the Reviewer is if there is a short-lived Ni(II) species to produce during UOR on Ni^{3+} active species. To discuss this possibility, we need to distinguish the resting state and working state for a given electrochemical reaction. Here, the working state means that there is a steady current flowing through the electrode, and resting state describes a steady state without electron transfer. Indeed, as pointed out by the Reviewer, under catalytic conditions, a working state for catalyst, the potential applied is always more positive than $E(\text{Ni(III/II)})$ to maintain the Ni^{3+} states. No complexing reaction occurs on NiOOH at the working state. However, we switched the working state to a resting state, that is, this electrode is under open-circuit conditions, the complexing reaction was occurring after spontaneous chemical reduction of NiOOH by urea to Ni(OH)_2 (**Supplementary Fig. 9**). This fact means that the dimethylglyoxime can chelate Ni^{2+} in the amorphous and defect-rich resting-state Ni(OH)_2 . Further, we carried out the CV cycling to simulate the working-state Ni(OH)_2 . And, during the CV cycling with a wide potential window from 0.9 to 1.3 V for Ni(OH)_2 generation, the complexing reaction occurs and completely etches the Ni(OH)_2 away from the electrode. These evidences suggested no occurrence of Ni(OH)_2 during UOR on NiOOH.

This is a truly key point, in our study, if there are short-lived Ni(II) species during UOR on NiOOH. To resolve this doubt, we would need to look back on the electron transfer mechanism of electrochemical reaction. In 1931, a quantum-mechanical theory of electron transfer at the electrode/electrolyte interface was originated by R. W. Gurney (*Proceedings of the Royal Society of London. Series A, Containing Papers of a Mathematical and Physical Character*, 1931, 134, 137-154). According to this theory, the steady current flowing through the electrode is a result of electron transitions between levels of equal energy at the electrode/electrolyte interface via tunneling the interfacial barriers. Subsequently, in 1952, W. F. Libby (*J. Phy. Chem.* 1952, 56, 863) introduced the Franck-Condon principle to describe the electrochemical electron transfer. The isoenergetic tunneling of electrons has been proved to occur when reducing and oxidizing ions or molecules satisfy the requirement of the Franck-Condon principle. An electron transfer in electrochemical reaction is like a charge-transfer transition in electronic spectroscopy. It occurs in a very short time and the various nuclei can be thought of as fixed in position during the transition (the Franck-Condon principle). Libby pointed out that when an electron is transferred from one ion or molecule to another, the “jump” is instantaneous and the nuclear environment around each ion does not have time to change during the jump itself (the Franck-Condon principle).

In 1955, R. A. Marcus developed an electron transfer theory, called Marcus theory (*J. Chem.*

Phys. 1956, 24, 966; *J. Chem. Phys.* 1965, 43, 679; *Ann. Rev. Phys. Chem.* 1964, 15, 155). In Marcus theory, the electron transfers in ordinary electrochemical reactions are adiabatic radiationless process, an isoenergetic transfer process between a position of Fermi level of the electrode and a position of equal energy in the discharging/or charging atom or molecule at the electrode surface. In the so-called outer-sphere approximation, the electron motion between its initial and final states was considered to be much more rapid than nuclear motions. Such an electronic motion, if considered a half-vibration, might take 10^{-15} s, whereas the time required for a proton to move to a new position would be 10^{-13} s, and 10^{-11} s for a heavy molecule. Thus all heavy nuclei were imagined as frozen in space during the electron motion.

Accordingly, according to the Marcus theory, the electron transfer during UOR on NiOOH is rapid process without electron accumulation to produce short-lived Ni^{2+} to satisfy the Franck-Condon principle. That is, if the short-lived Ni^{2+} can occur during UOR, the electron transfer will be not an adiabatic isoenergetic transfer process due to the different energy levels between Ni^{2+} and Ni^{3+} species.

Some secondary issues:

- Fig 4d. IR: C-N and C-O vibrations are almost baseline. Also, is there rationale why these vibrations are observed but no N-H ones?

Reply: Thank you very much for your help. The in situ electrochemical attenuated total reflection-surface enhanced infrared absorption spectroscopy (ATR-SEIRAS) was carried out to check the species evolution during UOR. As shown in **Fig. 5c**, deprotonation of amino group of urea first occurs to form -NH at 3442 cm^{-1} (*Angew. Chem. Int. Ed.* 2021, 60, 7297-7307) and 1645 cm^{-1} (*Angew. Chem.* 2019, 131, 16976-16981) as the potential increases. When potentials varied from 1.35 to 1.6 V, the vibration for CNO^- at 2168 cm^{-1} (*Angew. Chem.* 2019, 131, 16976-16981) appeared and gradually increased, indicating the UOR occurs once potential is above 1.3 V. During UOR, gradually increasing -NH₂ at 3250 cm^{-1} (*Angew. Chem. Int. Ed.* 2021, 60, 7297-7307), C=O at 1675 cm^{-1} (*Angew. Chem. Int. Ed.* 2021, 60, 7297-7307), and H₂O at 3642 cm^{-1} (*J. Am. Chem. Soc.* 1995, 117, 2118-2119) corresponded to adsorption enhancement of reactants, generation of intermediates, and accumulation of products, respectively. And C-N at 1464 cm^{-1} (*Angew. Chem. Int. Ed.* 2021, 60, 7297-7307) is slightly negative-going, indicating urea consumption. This result suggested that oxidation of urea undergoes the removal of protons in -NH₂, probably the most thermodynamic favorite process. Thus, considering thermodynamic favorite deprotonation of amino group of urea, N-H signals are hard to detect in ex situ FTIR.

- Sup. Fig. 13: the UOR equation is not equilibrated (O). I would at this pH write it with H₂O, not H⁺. And thus the Nernst equation should also be corrected.

Reply: Thank you very much for your help. The chemical equation of anodic reaction of UOR is $CO(NH_2)_2 + 6OH^- - 6e^- = N_2 + CO_2 + 5H_2O$. In theory, if the protons in the UOR are highly solvated to be released into the electrolyte, $\nu_{OH^-} = 6$ and $n = 6$. According to the Nernst equation, the equilibrium condition for this reaction under 25 °C and 1 atm is

$$\begin{aligned}\varphi &= \varphi^0 + \frac{RT}{nF} \ln \frac{a_{N_2} a_{CO_2} a_{H_2O}^5}{a_{CO(NH_2)_2} a_{OH^-}^6} \\ &= \varphi^0 - \frac{RT}{nF} \ln a_{OH^-}^6 \\ &= \varphi^0 - \frac{2.3RT}{6F} * 6 * \log a_{OH^-} \\ &= \varphi^0 - 0.059pH\end{aligned}$$

The pH dependence of UOR potential-current curves was recorded to understand the UOR mechanism (**Supplementary Fig. 14a**). To exclude the influence of background current and Ni(OH)₂/NiOOH oxidation current, we obtain the initial UOR potential by the threshold current density method to define a potential at 10 mA cm⁻² as UOR potential (V_{10 mA}). Here, the V_{10 mA} for UOR on Ni(OH)₂ electrode is a linear function of pH with a negative slope of -54 mV/pH (**Supplementary Fig. 14b**), much close to -59 mV/pH slope of theoretical V_{10 mA} versus pH for UOR with the completely solvated protons. Thus, the hydrogen of urea is released to electrolyte by the solvated reaction during UOR.

Supplementary Figure 14 | a, LSV curves for Ni(OH)₂ electrode in 0.33 M urea-containing electrolyte with different pH values. To avoid the conductivity difference caused by the ion concentration difference, K⁺ concentration was balanced to 1 M with the addition of K₂SO₄. **b**, The linear relationship of potential at 10 mA current vs. pH for Ni(OH)₂ electrode.

- An “onset” potential has no meaning. One should better report the potential at a given current density.

Reply: Thank you very much for your help. Here, the initial UOR potential is defined as a potential to reach a current density of 10 mA cm⁻², that is, a threshold current density method (*Curr. Opin. Electrochem.* 2023, 37, 101176) to determine the initial UOR potential at which the current of a Faradaic process is measurable.

- The text is often very hard to read and understand. An important style-editing would be needed before publication.

Reply: Thank you very much for your help. We have rechecked and revised the sentences in whole text that are possible to make confusing.

In conclusion, the main claim of the authors does not appear to be backed enough by data and there is at this stage no solid experimental evidence that sustains the mechanistic hypothesis. In my opinion, the strongest claim that can currently be made is that Ni(II) are not long-lived intermediates (/resting states), which is however not surprising given the applied potentials. If the authors wish to further address the implication of the Ni(III) and Ni(II) species in UOR, I would suggest performing additional experiments, such as with operando spectroscopies that enable addressing oxidation state (XAS, XPS,...). On these grounds, I have selected reject and resubmit.

Reply: Thank you very much for your help. We performed the in situ X-ray absorption near edge structure (XANES) measurements to monitor the Ni species during UOR as shown in Fig. 3d. The in situ XANES spectra revealed that the intensity of the white line decreases with polarizing the Ni(OH)₂ electrode from 1.3 to 1.5 V in 1 M KOH + 0.33 M urea, suggesting the formation of higher-valence Ni species due to that the high-valence Ni inducing distortion in the NiO₆ octahedral configuration will result in a decrease in the white line intensity (*J. Electrochem. Soc.* 1990,137, 383-388). Meanwhile, the edge energy (measured at half height) for the electrode at 1.3 and 1.5 V is the same as those of the Ni(OH)₂ (8342.6 eV) and NiOOH (8344.5 eV), respectively. This edge position suggests a +3 nickel oxidation state for the electrode during UOR (*J. Am. Chem. Soc.* 2012, 134, 6801-6809). The in situ XANES results confirmed that during UOR the Ni³⁺ is UOR-active and is likely to keep a constant valence state, in good agreement with the results from in situ Raman and in situ UV-vis absorption.

Fig. 3 | Evidences to trivalent nickel as active species for urea electrooxidation. d, Normalized in situ Ni K-edge XANES spectra for polarizing Ni(OH)₂ electrode at different potentials in 1 M KOH + 0.33 M urea, referenced to Ni foil and NiOOH.

In addition, in principle, as described in the mentioned-above discussions, we believe that the applied potentials polarize the Ni(OH)₂ to form NiOOH, and then NiOOH is the stable working state to isoenergetically transfer electrons at the UOR elementary reaction steps. As described by Marcus theory, for an electrochemical process with a steady current flowing through the electrode, the isoenergetic tunneling of electrons (adiabatic radiationless process) has been proved to occur when reducing and oxidizing ions or molecules satisfy the requirement of the Franck-Condon principle. At each elementary reaction step, the active centers of the electrode are expected to exhibit the stable architecture for creating isoenergetic massive electron transfer. Accordingly, according to the Marcus theory, the electron transfer during UOR on NiOOH is rapid process without electron accumulation to produce short-lived Ni²⁺ to satisfy the Franck-Condon principle. That is, if the short-lived Ni²⁺ can occur during UOR, the electron transfer will be not an adiabatic isoenergetic transfer process due to the different energy levels between Ni²⁺ and Ni³⁺ species. Indeed, the in situ Raman, in situ UV-vis absorption, in situ XAFS, and in situ EIS have demonstrated that the electron transfer is via Ni³⁺ without visible changes in oxidation valence.

REVIEWERS' COMMENTS

Reviewer #2 (Remarks to the Author):

The authors have addressed all my comments satisfactorily and I recommend acceptance of this work in Nature Communications.

Reviewer #4 (Remarks to the Author):

In the revision of their work, Yan S. and coll. addressed most of the reviewers' comments although in a partially satisfactory manner.

In particular, I still do not find convincing evidence ruling out the occurrence of Ni(II) and I find the statement too strong for the proofs provided. I though recognize that the authors have tamed down the statement now referring to Ni(OH)₂/NiOOH in the text.

Regarding electrolysis: do the authors observe CO₂/carbonates in ¹³C NMR? N₂ is an usual contaminant in headspace GC and one should be extremely careful when quantifying this gas.

It is even more questioning that the authors state: "The selectivity and faradaic efficiency of the gaseous product of UOR in N₂-saturated electrolyte containing 1 M KOH + 0.33 M urea were analyzed by gas chromatography (GC, 8860, Agilent, USA)." I do not understand how the quantification of N₂ product in a N₂-saturated electrolyte is made.

Response to the Reviewers' comments

Reviewer #4 (Remarks to the Author):

In the revision of their work, Yan S. and coll. addressed most of the reviewers' comments although in a partially satisfactory manner.

In particular, I still do not find **convincing evidence** ruling out the occurrence of Ni(II) and I find the statement too strong for the proofs provided. I though recognize that the authors have tamed down the statement now referring to Ni(OH)₂/NiOOH in the text.

Response: Thanks for your suggestions. To the best of our knowledge, there is a lack of a picosecond or femtosecond time-resolved tool to detect the valence state variation during electrochemical reaction. In our case, the in-situ UV-vis absorption can detect the millisecond-scale absorption of Ni³⁺ at 500 nm. No differences in the peak intensity were observed when we acquired the spectra at a given potential twice as shown in **Response Fig. 1**, suggesting that there is no occurrence of Ni²⁺ species with a millisecond-scale life during UOR. We also fitted the Nyquist plot by a typical Randle's circuit, which was composed of solution and catalyst-conductive substrate electrical connection resistance (R_s), electron transfer resistance, and constant phase angle element for the catalyst bulk/catalytic layer interface ($R_{ct,M}$, CPE_{M-T} , and CPE_{M-P}) and the catalytic layer/electrolyte interface ($R_{ct,UOR}$, CPE_{UOR-T} , and CPE_{UOR-P}), as shown in **Supplementary Fig. 12**. When the applied potential is higher than 1.30 V, a UOR potential, both the $R_{ct,M}$ and $R_{ct,UOR}$ are as low as 0.076 - 0.487 ohm, suggesting that the electron transfer during UOR on NiOOH is similar to the electron conduction in a metallic conductor. As described in our previous response, the Marcus theory tells us that for an electrochemical process with a steady current flowing through the electrode, the isoenergetic tunneling of electrons (adiabatic radiationless process) has been proved to occur when reducing and oxidizing ions or molecules satisfy the requirement of the Franck-Condon principle. At each elementary reaction step, the active centers of the electrode are expected to exhibit the stable architecture for creating isoenergetic massive electron transfer. Accordingly, according to the Marcus theory, the electron transfer during UOR on NiOOH is rapid process without electron accumulation to produce short-lived Ni²⁺ to satisfy the Franck-Condon principle. That is, if the short-lived Ni²⁺ can occur during UOR, the electron transfer will be not an adiabatic isoenergetic transfer process due to the different energy levels between Ni²⁺ and Ni³⁺ species. Therefore, we believe that the isoenergetic electron transfer during UOR on NiOOH does not undergo the short-lived Ni²⁺. The electron transfer process via the unoccupied orbitals of Ni³⁺ is similar to the electron transfer via a metallic conductive wire that transports electrons without energy loss if the resistance of circuit is not considered. That is, when applied a constant potential that can polarize the electrode to the energy level of Ni³⁺, the electrons from urea can be isoenergetically transferred into external circuit via unoccupied orbitals of Ni³⁺ with a transfer rate in picosecond to femtosecond scale. In other words, the electron transfer in electrochemical reaction is driven by a constant external potential, which is inherently different from the other electron-mediated reactions such as photocatalysis. In photocatalysis, we can observe the valence variation due to that there is no constant electric field to drive the electron transfer.

To provide an experimental evidence to show the electron transfer kinetics in the Ni³⁺ species and Ni³⁺ species with electron accumulation in a picosecond or femtosecond time scale, the femtosecond transient absorption spectra (fs-TAS) were carried out. According to the XPS results (**Supplementary Fig. 1d**), the adsorption of urea tends to stabilize the Ni(OH)₂ at low valence due to that the urea molecule with nucleophilic carbonyls is an electron donor. This means that the strong interactions between urea and Ni(OH)₂ change the surface electronic states of Ni(OH)₂. Obviously, the Ni species on the surface of urea-Ni(OH)₂ represent a lower valence state compared to Ni species on the surface of Ni(OH)₂. Accordingly, the adsorption of urea is likely to charge the surface of Ni(OH)₂ to form surface-charged Ni(OH)₂, thus slightly increasing the ground state of Ni(OH)₂. We found that the Ni(OH)₂ can be excited by 320 nm pump light and the light absorption of the excited-state Ni(OH)₂ is similar to that of Ni³⁺ with a characteristic absorption at 500 nm - 550 nm. As shown in **Supplementary Fig. 13a**, after light excitation at 320 nm, both the Ni(OH)₂ and urea-Ni(OH)₂ are excited from ground

state to an excited state. Therefore, the Ni in excited-state $\text{Ni}(\text{OH})_2$ and excited-state $\text{urea-Ni}(\text{OH})_2$ can be considered to be Ni^{3+} state and Ni^{3+} state with electron accumulation, respectively. That is, the decay of the excited-state $\text{Ni}(\text{OH})_2$ and $\text{urea-Ni}(\text{OH})_2$ is able to reflect the electron transfer kinetics in Ni^{3+} state and Ni^{3+} state with electron accumulation, respectively. We detected the decay kinetics from the excited state to ground state at 530 nm. As shown in **Supplementary Fig. 13b**, the decay time for the excited-state $\text{Ni}(\text{OH})_2$ and $\text{urea-Ni}(\text{OH})_2$ is 8 ps and 15 ps, respectively, indicating the charge transfer on Ni^{3+} state was faster than that on the Ni^{3+} state with charge accumulation. Thus, electrons tend to transfer through Ni^{3+} state directly, rather than accumulating to form transient Ni^{2+} state which hinders electron transfer.

Response Fig. 1 | In situ UV-vis absorption spectra of $\text{Ni}(\text{OH})_2$ electrode detected twice at 1.45 V.

Supplementary Figure 12 | Fitting analysis of EIS Nyquist plots. The data were from Supplementary Fig. 10 for $\text{Ni}(\text{OH})_2$ electrode at **a**, 1.30 V and **b**, 1.45 V. The insets show the equivalent circuit and the fitting data.

Supplementary Figure 1 | Ni(OH)₂ electrode. **d**, Ni 2p XPS spectra for the as-prepared Ni(OH)₂ and the Ni(OH)₂ after UOR.

Supplementary Figure 13 | Transient absorption spectra to check the electron transfer kinetics in Ni³⁺ state and Ni³⁺ state with electron accumulation. **a**, Schematic diagram of excited state and ground state. The adsorption of urea tends to stabilize the Ni(OH)₂ at low valence due to that the urea molecule with nucleophilic carbonyls is an electron donor. Accordingly, the adsorption of urea is likely to charge the surface of Ni(OH)₂ to form surface-charged Ni(OH)₂, thus slightly increasing the ground state of Ni(OH)₂. Therefore, the Ni in Ni(OH)₂ and urea-Ni(OH)₂ can be considered to be Ni³⁺ state and Ni³⁺ state with electron accumulation, respectively. That is, the decay of the excited-state Ni(OH)₂ and the excited-state urea-Ni(OH)₂ are able to reflect the electron transfer kinetics in Ni³⁺ state and Ni³⁺ state with electron accumulation, respectively. **b**, Femtosecond transient absorption spectra of Ni(OH)₂ and urea-Ni(OH)₂ detected by 530 nm probe light after excitation at 320 nm pump light.

Regarding electrolysis: do the authors observe CO₂/carbonates in ¹³C NMR?

Response: We did not observe the CO₂/carbonates in ¹³C NMR. This is because the CO₂ bubbles produced on the electrode tend to overflow from the electrolyte. Indeed, during UOR, a large number of bubbles form and leave the electrode surface. Therefore, the small amount of CO₂ was dissolved into electrolyte during the sample collection period, which is lower than the detection limitation of NMR.

N₂ is an usual contaminant in headspace GC and one should be extremely careful when quantifying this gas.

Response: In our case, we do not use a headspace GC to detect the gaseous products. The gas chromatography (GC, 8860, Agilent, USA) was used to detect the N₂ product. The products were collected in an airtight reactor and were directly introduced into sampling chamber of GC. Therefore, the generation and detection of the gas were carried out in an airtight system to avoid the introduction of the contaminants.

It is even more questioning that the authors state: “The selectivity and faradaic efficiency of the **gaseous product** of UOR in N₂-saturated electrolyte containing 1 M KOH + 0.33 M urea were analyzed by gas chromatography (GC, 8860, Agilent, USA).”

Response: We have revised this description as “*The selectivity and faradaic efficiency of the gaseous N₂ product of UOR in N₂-saturated electrolyte containing 1 M KOH + 0.33 M urea were analyzed by gas chromatography (GC, 8860, Agilent, USA)*”.

I do not understand how the quantification of N₂ product in a N₂-saturated electrolyte is made.

Response: To promote the N₂ bubbles to overflow from the electrolyte, the reactor before UOR was purged with N₂ for 10 min to achieve an N₂-saturated electrolyte. The quantification of N₂ product was based on the external standard method, according to the peak area detected by GC.